# Generation of human iPSC-derived phrenic-like motor neurons to model respiratory motor neuron degeneration in ALS
Louise Thiry[1,2], Julien Sirois[1,2], Thomas M. Durcan[1,2] & Stefano Stifani [1] ✉

The fatal motor neuron (MN) disease Amyotrophic Lateral Sclerosis (ALS) is characterized by progressive MN degeneration. Phrenic MNs (phMNs) controlling the activity of the diaphragm are prone to degeneration in ALS, leading to death by respiratory failure. Understanding of the mechanisms of phMN degeneration in ALS is limited, mainly because human experimental models to study phMNs are lacking. Here we describe a method enabling the derivation of phrenic-like MNs from human iPSCs (hiPSC-phMNs) within 30 days. This protocol uses an optimized combination of small molecules followed by cell-sorting based on a cell-surface protein enriched in hiPSC-phMNs, and is highly reproducible using several hiPSC lines. We show further that hiPSC-phMNs harbouring ALS-associated amplification of the *C9orf72* gene progressively lose their electrophysiological activity and undergo increased death compared to isogenic controls. These studies establish a previously unavailable protocol to generate human phMNs offering a disease-relevant system to study mechanisms of respiratory MN dysfunction.

Respiratory failure is the primary cause of death in ALS patients[1,2] and survival time is significantly shorter for patients with respiratory-onset ALS than bulbar- or limb-onset ALS[3,4], highlighting the importance of respiratory muscles in ALS patient survival and the crucial need to precisely understand the mechanisms underlying the respiratory pathology of the disease. The loss of the respiratory function of ALS patients is mainly the result of the degeneration of phMNs that control the major inspiratory muscle, the diaphragm[5]. Phrenic MNs are different from other spinal MNs in a number of ways, including their distinct developmental origin, topology, and electrophysiological properties[6–9]. Moreover, phMN progressive loss in ALS animal models begins before symptoms onset[10], and is more pronounced than other spinal respiratory MNs, such as hypoglossal (XII) MNs[11,12], reflecting the recognized selectivity of cell death mechanisms among different MN subtypes[5,13–16]. Progress in understanding the mechanisms of phMN degeneration in ALS has been hampered by the lack of adequate experimental systems to study human phMN (patho)physiology. Model systems based on other MN subtypes cannot adequately account for the specialized biology of phMNs and their vulnerability to degeneration in ALS.

The best characterized and most commonly used hiPSC-based MN derivation protocols generate heterogeneous cultures containing mainly MNs of the lateral (LMC) and median (MMC) motor columns[17–21]. Phrenic MNs are underrepresented in these cultures, with only 10% of MNs expressing the phMN marker gene, *SCIP*[22]. The intrinsic heterogeneity of currently available hiPSC-derived MN preparations, and the scarcity of phMNs in these cultures, make them inadequate to study phMN (patho) physiology. We therefore sought to develop methods to generate phMN-enriched cultures from hiPSCs.

During spinal cord development, phMNs emerge from specific MN progenitors (MNPs) located in the 'dorsal-most' MN progenitor (pMN) domain at cervical level[7,8,23,24]. Specification of the pMN domain is under the control of a ventral to dorsal gradient of Sonic hedgehog (SHH) signaling emanating from the notochord and floor plate[25–27]. Cervical identity is controlled by a rostro-caudal gradient of Retinoic Acid (RA), which regulates *HOXA5* gene expression in the cervical segment of the spinal cord, contributing to phMN identity specification[7,8,24]. Thus, we hypothesized that a calibrated activation of SHH and RA signaling in hiPSC-derived neural

[1]Department of Neurology and Neurosurgery, Montreal Neurological Institute-Hospital, McGill University, 3801, rue University, Montreal, QC H3A 2B4, Canada. [2]Early Drug Discovery Unit, Montreal Neurological Institute-Hospital, McGill University, 3801, rue University, Montreal, QC H3A 2B4, Canada. ✉e-mail: stefano.stifani@mcgill.ca

progenitor cells (NPCs) might provide a strategy to enhance the specification of dorsal MNPs with cervical identity, in turn generating neurons with phMN features.

Here we show that exposure of hiPSC-derived NPCs to optimized concentrations of RA and the SHH agonist purmorphamine, followed by Fluorescence Activated Cell Sorting (FACS) of MNPs based on the cell surface protein IGDCC3, yields cultures highly enriched in MNs expressing typical phMN markers (hiPSC-phMNs). We show further that hiPSC-phMNs obtained from hiPSC lines derived from ALS patients with *C9orf72* amplification, the most common familial ALS mutation, provide a disease-relevant experimental system to address existing ambiguities about the impact of *C9orf72* amplification on MN survival[28–31] and activity[28,32,33]. Specifically, *C9orf72*-mutated hiPSC-phMNs are hypoactive and undergo increased death compared to isogenic controls. These results provide a method to generate human phMNs offering a disease-relevant model system to study mechanisms of respiratory MN dysfunction.

## Results

### Specification of phrenic motor neuron progenitors from human iPSCs

Respiratory MNs are generated from "dorsal-most" MNPs in response to a combination of SHH[27] and RA[7,8] signaling gradients, and are characterized by lower expression of the transcription factor TLE and higher expression of the transcription factor PAX6 compared to "ventral-most" MNPs[34]. We therefore hypothesized that controlled activation of SHH signaling to promote the specification of hiPSC-MNPs (MNPs hereafter) displaying a PAX6[HIGH]/TLE[LOW] expression profile ("dorsal-most"), in conjunction with RA concentrations imparting a cervical identity, would enhance the generation of MNPs competent to give rise to phMNs. To test this hypothesis, NPCs obtained from the hiPSC line NCRM1 were ventralized through the addition of the SHH agonist purmorphamine (Pur.), used at either 0.5 μM or 0.125 μM, in the presence of different concentrations of RA (0.1 μM or 1 μM) (Fig. 1a). We shall hereafter define as *generic* condition the combination of 0.1 μM RA + 0.5 μM Pur., which has been used in many previous hiPSC-MN differentiation studies[18,21,22,32,35]. After 12 days of differentiation, all four culture conditions defined by the different combinations of RA and Pur. resulted in the generation of OLIG2[+] MNPs in similar proportions (Fig. 1b, c). Analysis of the relative levels of fluorescence intensity of TLE and PAX6 staining in each cell allowed for the identification of their PAX6[HIGH]/TLE[LOW] ("dorsal-most") expression profile (Supplementary Fig. 1). Importantly, increased RA (1 μM) and reduced purmorphamine (0.125 μM) (RA[HIGH]/Pur.[LOW]) resulted in a 3-fold enrichment in OLIG2[+]/TLE[LOW]/PAX6[HIGH] (dorsal-most) MNPs when compared to the *generic* protocol (Fig. 1d). Variation of either RA or purmorphamine alone had no significant effect on the proportion of the dorsal-most MNPs. As expected, the *generic* method also induced a small proportion (1.8 ± 0.3%) of NKX2.2[+] V3 ventral interneuron progenitors (INPs), which are located just ventral to the pMN domain in the developing spinal cord in vivo[25,26]. The RA[HIGH]/Pur.[LOW] condition led to a 10-fold reduction in the proportion of NKX2.2[+] V3 INPs, further pointing to a dorsalization of the progenitor cell pool compared to the *generic* protocol (Supplementary Fig. 2a, b).

The combined signals from graded SHH and RA are integrated primarily by the HOX transcription factors to specify rostro-caudal MN subtype identity[24]. Moreover, phMN identity is specified by the expression of HOXA5 in the cervical segment of the spinal cord[7,8]. We therefore quantified the proportion of HOXA5-expressing MNPs generated under the four culture conditions (Fig. 1e, f). A roughly 2-fold enrichment in cervical HOXA5[+] MNPs was induced in the RA[HIGH]/Pur.[LOW] condition compared to the other 3 conditions (Fig. 1f). Most of HOXA5[+] MNPs were also PAX6[HIGH]/TLE[LOW] ("dorsal-most"), confirming the specification of dorsal MNPs with cervical identity (Supplementary Fig. 2c). Together, these results provide evidence that modulation of RA and SHH signaling can be used to promote the generation of cells with properties of phMN progenitors from hiPSCs.

### Differentiation of phrenic-like motor neurons from human iPSC-derived cervical dorsal-most motor neuron progenitors

Continued exposure of MNPs to RA and Pur. led to the generation of induced cells expressing the typical pan-MN markers ISL1 and HB9[18] after 25 days of differentiation in all four culture conditions described above (Fig. 2a, b). The majority of ISL1[+]/HB9[+] hiPSC-MNs (MNs hereafter) derived under the *generic* condition, or with varying concentrations of either RA or purmorphamine alone, expressed FOXP1, a marker of limb inner-vating MNs of the LMC[9,18,36]. Under those three conditions, a small proportion of the induced MNs were negative for FOXP1 and positive for LHX3, a marker of MMC MNs[37]. In addition, less than 10% of MNs were LHX3-negative and expressed the transcription factor SCIP, known to be enriched in vivo in phMNs[7,8,38] (Fig. 2c). Importantly, differentiation under the RA[HIGH]/Pur.[LOW] condition led to a significant increase in the proportion of LHX3[−]/SCIP[+] hiPSC-phMN-like cells (phMNs hereafter) (25.6 ± 4.2%), with a concomitant reduction in the proportion of FOXP1[+]/LHX3[−] LMC MNs (28.1 ± 1.2%), when compared to *generic* MN cultures (7.7±0.7% of phMNs; 51.4 ± 5.1% of LMC MNs). SCIP[+] phMNs were also HOXA5[+] and FOXP1-negative (Supplementary Fig. 3), further defining their phMN molecular identity[7,8]. Together, these observations show that exposure to defined concentrations of RA and purmorphamine can direct the differentiation of MNPs into cultures enriched in MNs exhibiting a phMN molecular profile in vitro (hereafter termed *phMN[ENRICHED]*).

### Characterization of enriched human phrenic-like motor neuron cultures by single cell RNA sequencing

To further compare the cellular composition of MN cultures differentiated under the *generic* or *phMN[ENRICHED]* condition, we performed microdroplet-based sc-RNAseq[39] followed by principal-component and t-distributed stochastic neighbor embedding (t-SNE) analysis (Supplementary Fig. 4a). Based on differential gene expression between each cell, 7 major cell types were identified in similar proportions in the *generic* and *phMN[ENRICHED]* conditions (Supplementary Fig. 4b, c).

As expected, the largest proportion of cells in both conditions was identified as MNPs or MNs. The 3,461 quality-controlled cells belonging to these categories were divided into 11 shared nearest neighbour graph-based clusters, suggesting heterogeneity in the MNP/MN population (Fig. 3a, b). Based on differential gene enrichment, 6 major types of cells were identified in the *generic* and *phMN[ENRICHED]* cultures (Fig. 3c). We observed very small proportions of apoptotic MNs with higher expression of *BAX*, *CASP3*, *NGF* and *FAS*[40], MNPs with higher expression of *NKX6.1*, *OLIG2* and *LHX3*[41], and LHX3[−]/FOXP1[+] MMC MNs. In both conditions, we also found similar proportions of MNs expressing the brainstem marker genes *OTX2* and *MAFB*: these cells were defined as 'incompletely caudalized'[42,43].

The largest proportion (53%) of MNs differentiated under the *generic* condition corresponded to FOXP1[+]/LHX3[−] LMC MNs. Importantly, LMC MNs represented only 27% of the MN population generated under the *phMN[ENRICHED]* condition, whereas the proportion of FOXP1[−]/LHX3[−]/HOXA5[+] cervical MNs was more than doubled (46%) in the latter compared to the *generic* condition (21%). In both cultures, some cervical FOXP1[−]/LHX3[−]/HOXA5[+] MNs were identified as phMNs because they expressed the phMN marker genes *SCIP* and *TSHZ1*[44], whereas the rest was annotated as 'immature phMNs'. Interestingly, while most clusters were equally represented in both cultures, some appeared be to enriched in the *generic* (cluster #4) or the *phMN[ENRICHED]* (cluster #6) condition (Fig. 3d). Differential gene enrichment analysis between each cluster showed that FOXP1[−]/LHX3[−]/HOXA5[+] MNs were mainly contained in cluster #6 (Fig. 3e; 35%) and enriched in the *phMN[ENRICHED]* condition (Fig. 3f). In contrast, the *generic*-specific cluster #4 contained only 5% of FOXP1[−]/LHX3[−]/HOXA5[+] MNs (Fig. 3e). Conversely, LMC MNs were found in lower proportions in cluster #6 (Fig. 3g, h; 15%) and were mainly contained in cluster #4 (28%). These observations confirmed a > 2-fold enrichment in phMNs obtained from hiPSCs under the *phMN[ENRICHED]* condition.

The smallest proportions of cells found in both cultures were identified as undifferentiated stem-like cells, NPCs, oligodendrocytes, and apoptotic cells.

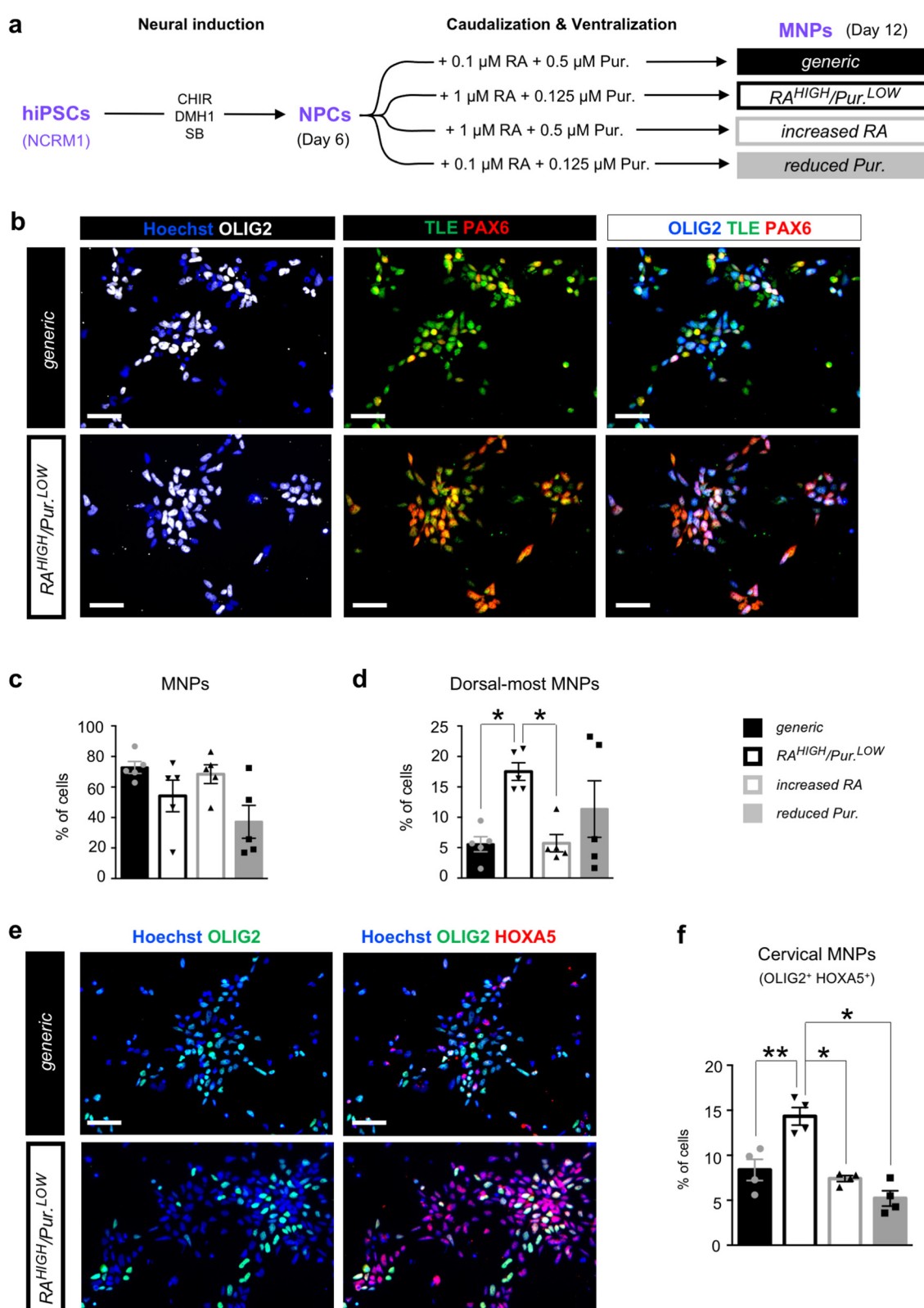

We also observed astrocytic glia and spinal IN subtypes. Interestingly, the majority of INs induced in the *generic* condition were the ventral V2 INs (*SOX14*[+]/*CHX10*[+]/*GATA3*[+]), while more dorsal IN populations (dINs; *LBX1*[+]/*TLX3*[+], or *LBX1*[+]/*PAX2*[+]) were the most represented in *phMN*[ENRICHED] condition, thus confirming the dorsalization of the latter culture (Supplementary Fig. 4d, e).

We next asked whether the *phMN*[ENRICHED] condition could direct phMN differentiation from other hiPSC lines. In particular, we sought to validate this protocol in ALS-patient derived hiPSC lines to provide an enhanced model system to study the specific susceptibility of phMNs to death in ALS. Two ALS-patient derived hiPSC lines carrying $G_4C_2$ repeat expansions in the *C9orf72* gene, the most frequently mutated gene in

**Fig. 1 | Characterization of enriched human iPSC-derived cervical dorsal-most motor neuron progenitors. a** Representation of the experimental design to generate hiPSC-derived motor neuron progenitors (MNPs) from neural progenitor cells (NPCs) induced using the indicated combination of small molecules. NPCs were treated at differentiation day 6 with four distinct combinations of retinoic acid (RA) and Purmorphamine (Pur.): 1) *generic* (filled black bar, 0.1 μM RA + 0.5 μM Pur.); 2) $RA^{HIGH}/Pur.^{LOW}$ (hollow black bar, 1 μM RA + 0.125 μM Pur.); 3) *increased RA* (hollow grey bar, 0.1 μM RA + 0.5 μM Pur.); and 4) *reduced Pur.* (filled grey bar, 0.1 μM RA + 0.125 μM Pur.). **b** Representative images of MNPCs stained with anti-OLIG2 (white), anti-TLE (green) and anti-PAX6 (red) antibodies after 12 days of differentiation under the *generic* or $RA^{HIGH}/Pur.^{LOW}$ conditions. **c, d** Quantification of all MNPs ((**c**); OLIG2$^+$ cells) and the dorsal-most MNPs ((**d**); PAX6$^{HIGH}$/TLE$^{LOW}$/OLIG2$^+$ cells) as percentages of the total number of cells derived in each of the four experimental conditions described in (**a**). **e** Representative images of MNPs stained with anti-OLIG2 and anti-HOXA5 antibodies after 12 days of differentiation under the *generic* or $RA^{HIGH}/Pur.^{LOW}$ conditions. **f** Quantification of cervical MNPs (HOXA5$^+$/OLIG2$^+$ cells) as percentages of the total number of cells derived in each of the four experimental conditions described in (**a**). Scale bars = 50 μm. One-way ANOVA and Holm Sidak's post-hoc multiple comparisons test; * = $p < 0.05$; $N = 4–5$ biologically independent cultures per condition (with > 500 cells in 3 random fields for each culture). Error bars are means ± standard error of means (SEM) of the average.

familial forms of ALS, and their matching isogenic controls were used. Hereafter, these hiPSCs will be referred to as CS29 and CS52 lines (see Methods for details). Similar to what we observed with NCRM-1 hiPSCs, the $phMN^{ENRICHED}$ condition resulted in a roughly 3-fold enrichment in SCIP$^+$ phMNs derived from ALS-patient and isogenic controls of both CS29 and CS52 lines (Supplementary Fig. 5f), demonstrating the applicability of this method to multiple hiPSC lines.

### Isolation of highly enriched populations of phrenic-like motor neurons

In order to increase the enrichment in phMNs, we sought to identify cell-surface markers that could be used to isolate phMNs from $phMN^{ENRICHED}$ cultures by FACS (Fig. 4). Based on differential gene enrichment between clusters #6 (i.e., cervical *FOXP1$^-$/LHX3$^-$/HOXA5$^+$* MNs) and #4 (i.e., LMC MNs), we identified the most differentially expressed genes (DEGs) up-regulated in phMNs compared to LMC MNs (Fig. 4a). The cell surface protein-coding gene *IGDCC3* was identified among the 10 most DEGs ($p = 9.18^{-173}$) over-expressed in phMNs (LogFC = 12.977). IGDCC3-positive cells were thus isolated by FACS from the already partially enriched MNP cultures obtained from the CS29 and CS52 lines differentiated for 18 days with the $phMN^{ENRICHED}$ condition. Regardless of their genetic background, approximately 15% of the live single-cells submitted to this stringent technique were identified as IGDCC3$^+$ cells: these cells were isolated from the culture and plated for expansion (Fig. 4b). One-week post-plating, a large proportion of the ISL1$^+$/HB9$^+$ MNs contained in the post-FACS cultures were identified as SCIP$^+$/LHX3$^-$ phMNs (Fig. 4c, d; 61.1 ± 1% for isogenic; 49.9±8.7% for ALS), suggesting a 2.5-fold enrichment compared to $phMN^{ENRICHED}$ cultures and a 5-fold enrichment compared to *generic* cultures. Following another week of differentiation in the $phMN^{ENRICHED}$ condition, post-FACS cultures were enriched in mature CHAT$^+$ MNs (Fig. 4e), among which 60% were SCIP$^+$/LHX3$^-$ phMNs (Fig. 4f), confirming a significant enrichment ($phMN^{ISOLATED}$ cultures hereafter). Post-FACS SCIP$^+$ cells were also FOXP1-negative and HOXA5$^+$ (Supplementary Fig. 6), further confirming their phMN molecular identity. Collectively, these results show that the $phMN^{ISOLATED}$ differentiation method provides a previously unavailable protocol to generate cultures highly enriched in phMNs using different hiPSCs lines.

### Study of respiratory motor neuron death in ALS using human iPSC-derived cultures enriched in phrenic motor neurons

To validate hiPSC-derived phMN cultures as a disease-relevant experimental system to study respiratory MN (patho)physiology, we next investigated two open questions in the field of ALS *C9orf72* research using these cell cultures. More specifically, it has been hypothesized that previous inconsistent results on the impact of *C9orf72* amplification on MN death[28–31] and spontaneous MN activity[28,32,33] might have been caused, at least in part, by the use of *generic* MN cultures with heterogeneous cellular compositions. Since changes in MN electrical activity are not only signs of altered electrophysiological properties, but also impact on MN survival[45–47], we first sought to compare the consequence of *C9orf72* mutation on both MN activity and survival using *generic* and $phMN^{ENRICHED}$ cultures.

We quantitated the proportion of MNs expressing the apoptotic marker cleaved-caspase-3 (CC3) in *C9orf72*-mutated (ALS) and isogenic MNs over the course of several weeks in vitro. Significant difference between isogenic and ALS MNs was only detected after 7-weeks of maturation in the *generic* condition (Fig. 5a, b) (although not statistically significant in the case of the CS52 line (Supplementary Fig. 7a)). In contrast, when cells were differentiated using the $phMN^{ENRICHED}$ condition, the proportion of dying MNs was higher in ALS compared to isogenic cultures already at 4 weeks post-plating, with both the CS29 (Fig. 5a, b) and CS52 (Supplementary Fig. 7a) lines. After 5 weeks of maturation in either culture conditions, quantification of the amount of ATP, which is directly proportional to the number of viable cells in culture, suggested reduced cell viability of ALS MNs compared to isogenic, in both the CS29 (Fig. 5c) and CS52 (Supplementary Fig. 7b) lines. Collectively, these results suggest that $phMN^{ENRICHED}$ cultures can reveal survival differences between *C9orf72*-muted and isogenic MNs that may be missed using *generic* preparations unless the latter are studied after prolonged times in culture.

We next compared spontaneous firing of *generic* and $phMN^{ENRICHED}$ isogenic and ALS MN cultures over time using extracellular multi-electrode arrays (MEA; Fig. 5d–g). As illustrated by the raster plots (Fig. 5d), the number of detected spikes (Fig. 5e), mean firing rate (Fig. 5f) and burst frequency (Fig. 5g) were significantly reduced in ALS compared to isogenic $phMN^{ENRICHED}$ cultures from CS29 iPSCs, suggesting a loss of activity as early as 4 weeks post-plating. Hypoexcitability with reduced number of spikes was also observed 4 weeks post-plating in *generic* and $phMN^{ENRICHED}$ cultures with the CS52 line (Supplementary Fig. 7c). Conversely, *generic* ALS MNs derived from the CS29 line exhibited increased number of spikes from 2 to 9 weeks post-plating (Fig. 5e), without significant difference in the weighted mean firing rate or burst frequency (Fig. 5f, g). Application of GABAergic and glycinergic transmission blockers did not impact firing (Supplementary Fig. 8), suggesting that activity of inhibitory neurons was minimal and that the MEA recordings reflected intrinsic changes in MN excitability.

Since conflicting results were observed using *generic* and $phMN^{ENRICHED}$ MN cultures derived from CS29 hiPSCs, we next focused on this specific line to further compare the electrical activity and survival of isogenic and ALS $phMN^{ISOLATED}$ preparations containing a higher fraction of phMNs (Fig. 6). As early as 5 weeks post-plating, the number of spikes, mean firing rate and burst frequency were significantly reduced in ALS $phMN^{ISOLATED}$ preparations, suggesting a progressive loss of activity compared to isogenic controls (Fig. 6a–c). This was correlated with a dramatic increase in the proportion of CC3$^+$ apoptotic MNs in ALS $phMN^{ISOLATED}$ cultures compared to isogenic, as of 4-weeks post-plating (Fig. 6d, e).

Collectively, these studies show that hiPSC-derived MN cultures enriched in phMNs can consistently reveal disease-relevant phenotypes in multiple lines, whereas *generic* MN preparations exhibit either non-significant or inconsistent phenotypes under equivalent experimental conditions. These observations suggest that MN preparations enriched in phMNs provide an improved model system to study changes in survival and electrophysiology in respiratory MNs highly susceptible to degeneration in ALS.

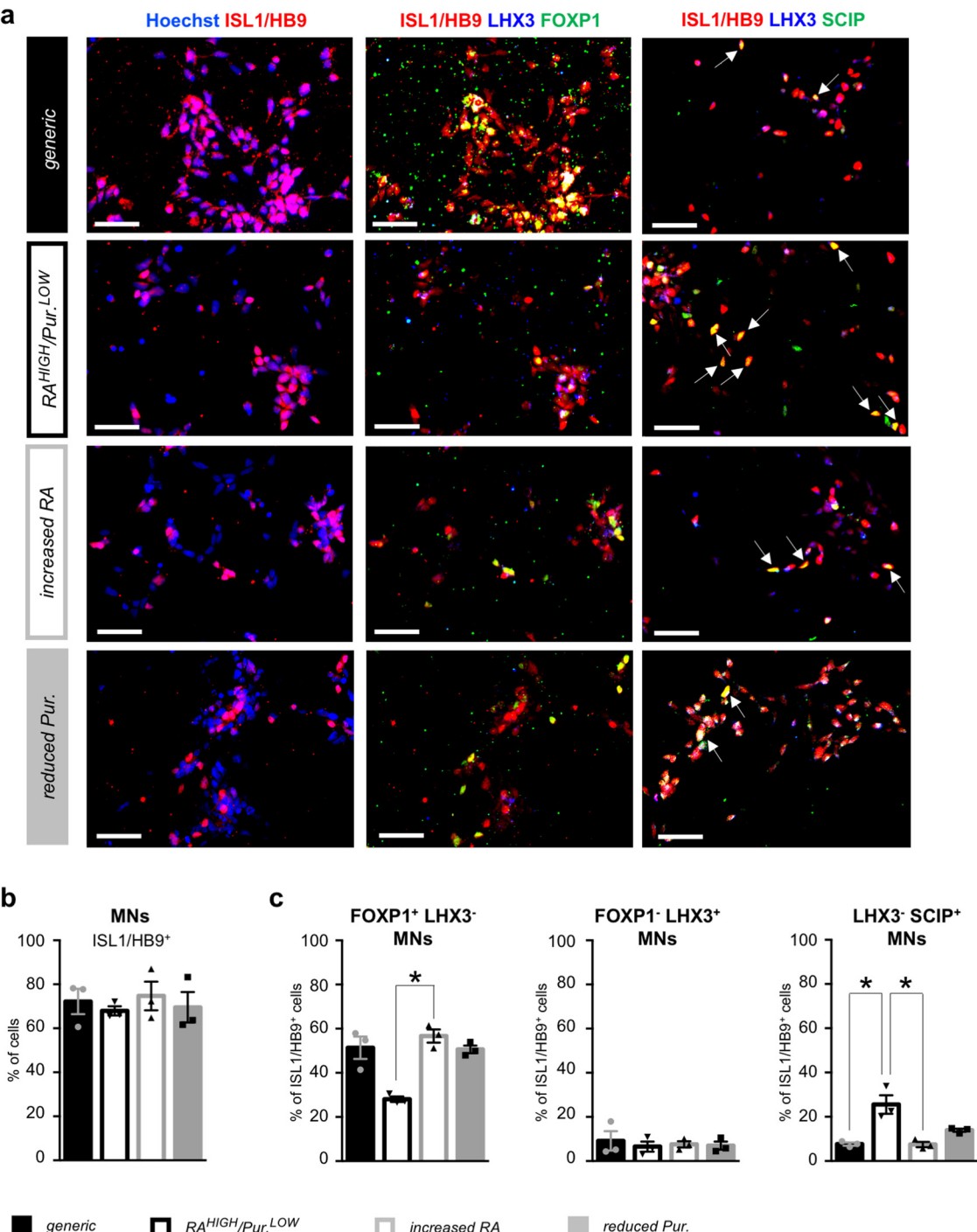

**Fig. 2 | Characterization of enriched human iPSC-derived phrenic motor neurons by immunocytochemistry. a** Representative images of motor neurons (MNs) co-stained with antibodies against the known pan-MN marker HB9/ISL1 (red), LHX3 (blue), and FOXP1 or SCIP (green) after 25 days of differentiation under the four experimental conditions defined in Fig. 1a. Arrows indicate phMNs, identified as HB9/ISL1$^+$ LHX3$^-$ SCIP$^+$. **b** Quantification of ISL1$^+$/HB9$^+$ MNs, as percentages of the total number of cells. **c** Quantification of LMC MNs identified as FOXP1$^+$/

LHX3$^-$, MMC MNs identified as LHX3$^+$/FOXP1$^-$, and phMNs identified as LHX3$^-$/SCIP$^+$, as percentages of the number of ISL1$^+$/HB9$^+$ MNs. Scale bars = 50 μm. MMC = median motor column, LMC = lateral motor column. Friedman non-parametric test and Dunn's post-hoc multiple comparisons test; * = $p < 0.05$; $N = 3$ biologically independent cultures (with > 500 cells in random fields for each culture). Error bars are means ± standard error of means (SEM) of the average.

## Discussion

Respiratory function significantly predicts the survival and quality of life of patients with MN disease such as ALS, and respiratory failure following phMN degeneration is a major contributor to death in ALS patients. Understanding the mechanisms underlying the degeneration of phMNs is therefore of great importance in ALS research. However, most ALS hiPSC-

based studies rely on differentiation protocols that give rise to heterogenous MN cultures, with generally low yields of phMNs, varying or undetermined proportions of other MN subtypes, and variable efficiency between methods and from one hiPSC line to another[17–22,48–50]. The heterogeneity and limited characterization of hiPSC-derived MNs generated using different experimental methods may explain, at least in part, why studies using these

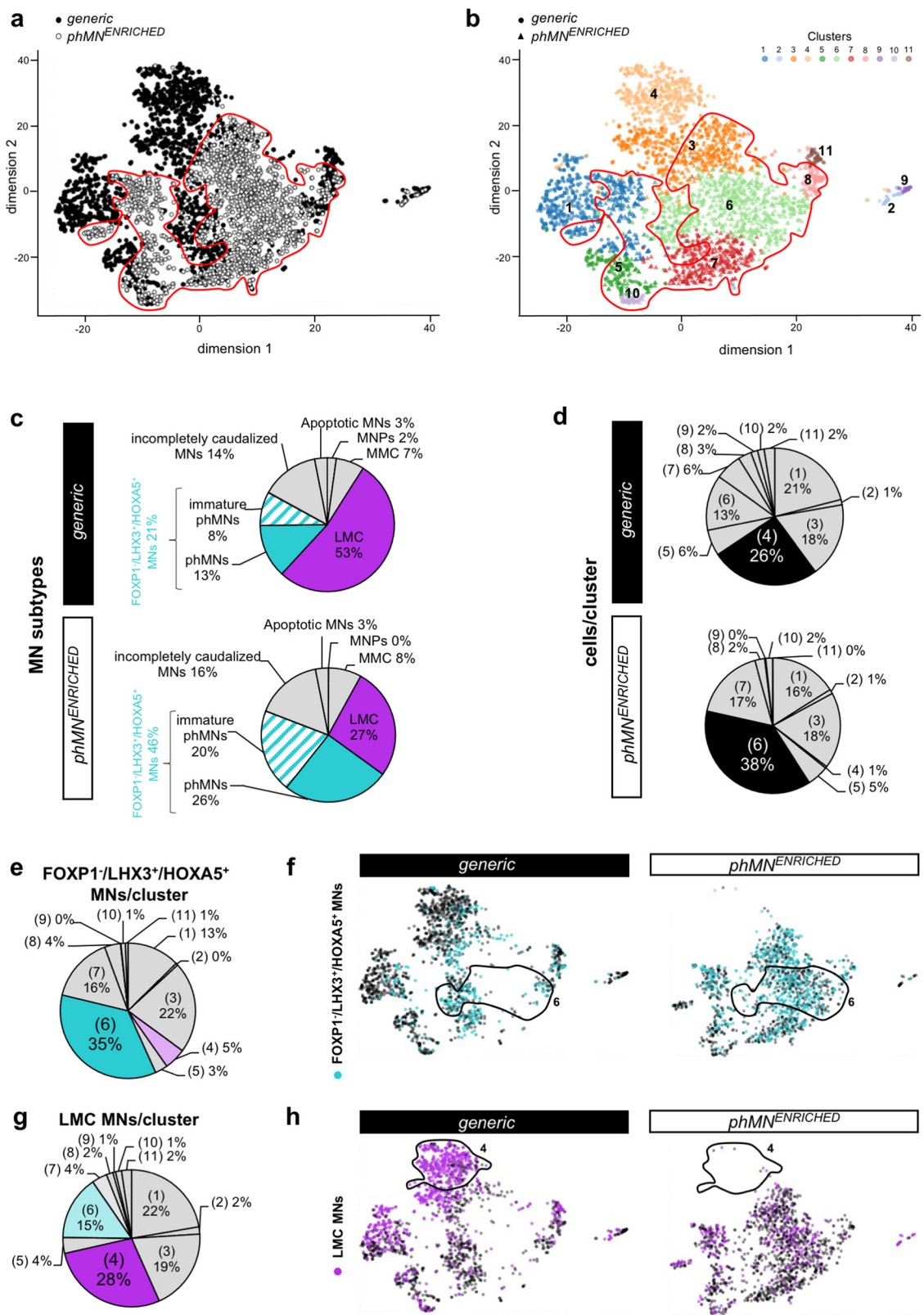

experimental systems for research into ALS pathophysiology have provided ambiguous results[28–33,35,51,52]. Based on these considerations, and in particular the low fraction of phMNs present in common hiPSC-derived MN preparations[22], the present study aimed at developing a method that would provide a robust source of human phMNs, and at validating the potential of

hiPSC-derived phMNs as a model to study respiratory MN degeneration in ALS.

Consistent with previous reports, our study revealed that a widely used (*generic*) MN differentiation protocol generates mainly LMC MNs, together with smaller populations of MMC MNs and phMNs[19,22,48,53]. Using this

**Fig. 3 | Characterization of enriched human iPSC-derived phrenic motor neurons by single cell RNA sequencing. a** t-SNE plot of all cells identified as MNs in hiPSC-derived MN cultures after 28 days of differentiation under either the *generic* (black dots) or *phMN*$^{ENRICHED}$ condition (white dots and circled area). **b** t-SNE plot of the same cells shown in (**a**) differentiated under either the *generic* (dots) or *phMN*$^{ENRICHED}$ condition (triangles), showing 9 shared nearest neighbour graph-based cell clusters. **c** Proportion of each MN subtype found in each of the indicated conditions, identified by their expression of specific genes: MNPs, *OLIG2*$^+$; LMC MNs, *FOXP1*$^+$/*LHX3*$^-$; MMC MNs, *FOXP1*$^-$/*LHX3*$^+$; phMNs, *FOXP1*$^-$/*LHX3*$^-$/ *HOXA5*$^+$ MNs expressing *SCIP* and/or *TSHZ1*; immature phMNs, *FOXP1*$^-$/*LHX3*$^-$/ *HOXA5*$^+$ MNs lacking the expression of *SCIP* and *TSHZ1*; incompletely caudalized MNs, *MAFB*$^+$, HOXB1$^+$. **d** Percentage of total cells differentiated under the *generic* (top pie chart) or *phMN*$^{ENRICHED}$ condition (bottom pie chart) present in each cluster. For both conditions, the cluster containing the majority of the cells is represented in black. **e** Distribution of all *FOXP1*$^-$/*LHX3*$^-$/*HOXA5*$^+$ MNs, identified as *FOXP1*$^-$/ *LHX3*$^-$/*HOXA5*$^+$, in the 9 clusters. **f** t-SNE plot of each indicated condition colored for *FOXP1*$^-$/*LHX3*$^-$/*HOXA5*$^+$ MNs (blue dots). **g** Distribution of all LMC MNs, identified as *FOXP1*$^+$/*LHX3*$^-$, in the 9 clusters. **h** t-SNE plot of each indicated condition colored for LMC MNs (purple dots). MMC Median motor column, LMC Lateral motor column.

differentiation method, we observed that ~30% of the cervical *FOXP1*$^-$/ *LHX3*$^-$/*HOXA5*$^+$ MNs identified by sc-RNAseq did not express the phMN markers *SCIP* and *TSHZ1* even after prolonged culture, suggesting the presence of developmentally immature phMNs. In possible agreement with this possibility, immunocytochemistry revealed a smaller proportion of SCIP$^+$ phMNs after 25 days of differentiation than the population detected by sc-RNAseq in more mature cultures. These observations suggest that phMNs differentiate in low numbers, and at a slow rate, using a *generic* hiPSC-MN derivation protocol. We therefore sought to develop a previously unavailable method to generate cultures enriched in phMNs from hiPSCs.

During spinal cord development, phMNs derive from progenitors located in the dorsal-most part of the pMN domain at cervical level[7,8,26,27,34]. We therefore aimed at developing in vitro culture conditions that would instruct MNPs to acquire molecular characteristics expected of dorsal pMN progenitors at cervical level, such as a HOXA5$^+$ (cervical trait) and Pax6$^{HIGH}$/ TLE$^{LOW}$ (dorsal pMN trait) expression profile. Testing a number of specific concentrations of RA and purmorphamine, we defined experimental conditions (*phMN*$^{ENRICHED}$) resulting in a 2-3-fold enrichment in both HOXA5$^+$ and Pax6$^{HIGH}$/TLE$^{LOW}$ MNPs, when compared to *generic* MN differentiation conditions. Consistently, upon further differentiation, *phMN*$^{ENRICHED}$ cultures exhibited a roughly 3-fold enrichment in SCIP$^+$ MNs, compared to *generic* cultures. Of note, the relative proportion of induced phMNs appeared lower in scRNA-seq studies compared to ICC. This could reflect a specific loss of phMNs over other MN types during single-cells suspension preparation prior to sequencing. Mature hiPSC-derived MNs tend to coalesce into large clusters[17,32,48,54–57], and phMN present a distinctive tight clustering very early during in vivo development[58,59]. The presence of these clusters necessitates harsh conditions to obtain single-cell suspensions for sc-RNAseq[22], and it is possible that phMNs are particularly sensitive to these conditions.

To further increase the proportion of phMNs in our preparations, we harnessed the sc-RNAseq data generated using these partially enriched cultures to identify cell surface markers that could be used to isolate phMNs by FACS. *IGDCC3* was identified as a gene encoding a cell surface protein among the 10 most DEGs up-regulated in the *phMN*$^{ENRICHED}$ cluster compared to the LMC MNs cluster. *IGDCC3* is expressed in the mouse developing nervous system, including spinal MNs. In the spinal cord, *IGDCC3* is down-regulated in maturing MNs[60,61]. Thus, although *IGDCC3* expression is not limited to phMNs, its expression in different MN subtypes might have different dynamics resulting in an apparent up-regulation in phMNs compared to LMC at a given time in development. In line with this hypothesis, post-IGDCC3 FACS cultures contained approximately 60% mature CHAT$^+$ MNs, among which ~60% were SCIP$^+$ (as well as HOXA5$^+$, FOXP1$^-$ and LHX3$^-$), demonstrating a significant enrichment in phMNs compared to *generic* conditions. In summary, the described method provides a simple and scalable approach to achieve a remarkable level of phMN enrichment compared to prior MN derivation protocols. It also holds potential for future enhancement thanks to the possibility of benefitting from single cell analyses of FACS-sorted cultures to identify additional cell surface markers that could be used in FACS-based strategies to obtain highly enriched populations of phMNs.

Importantly, we showed that phMN-enriched cultures provide a disease-relevant model to study ALS pathophysiology, offering advantages over *generic* MN preparations. Our studies showed that significant differences in MN activity and survival between ALS (*C9orf72*-mutated) and isogenic preparations could be detected as early as 4-weeks post-plating using *phMN*$^{ENRICHED}$ cultures. More specifically, ALS *phMN*$^{ENRICHED}$ preparations derived from two separate *C9orf72*-mutated hiPSC lines exhibited consistent loss of electrical activity and progressive MN death compared to isogenic counterparts. In contrast, studies with *generic* MN cultures resulted in variable and inconclusive results from one hiPSC line to the other. Ambiguities in the impact of *C9orf72* expansion on MN survival and excitability had emerged from previous studies relying on *generic* hiPSC-derived MNs. Some studies observed survival differences between ALS and control cells[29,30], while other studies did not[28,31]. Moreover, a spectrum of altered excitability was reported in ALS compared to control MNs, including hyperexcitability[33], loss of electrical activity[32], or transient hyperexcitability[28,35]. Those conflicting findings may have been a consequence of the use of inconsistent, and likely sub-optimal, experimental systems because of the variable composition of MN subtypes present in *generic* MN cultures. Consistent with this possibility, the ALS-related phenotypes observed using *phMN*$^{ENRICHED}$ preparations became more prominent in our hands after further enrichment following FACS-isolation of phMNs, with robust cell death and sustained loss of activity of ALS cells observed 5 weeks post-plating when studying *phMN*$^{ISOLATED}$ preparations. In contrast, we observed that ALS *generic* MN cultures showed either hyperexcitability or no excitability changes at the same time-point, similar to variabilities found in previous studies[28,35,62]. The latter observation could be explained, at least in part, by the finding that isogenic *phMN*$^{ENRICHED}$ cultures were significantly more active than isogenic *generic* preparations, thereby facilitating the detection of loss of activity in ALS cultures enriched in phMNs. The intrinsic hyperactivity of isogenic cultures enriched in phMNs, compared to *generic* preparations, may reflect the specific ability of phMNs to spontaneously fire very early during in vivo development[59], thus further supporting the physiological relevance of hiPSC-derived phMN preparations to study excitability changes in respiratory MN diseases.

In conclusion, the present work provides a previously unavailable method to generate MN cultures enriched in human phMNs from iPSCs. Our results demonstrate that these preparations offer a disease-relevant model system to study the mechanisms underlying the respiratory pathology of ALS. Specifically, these cultures can be studied for at least 10 weeks and disease-related phenotypes including cell death and loss of electrophysiological activity are consistently observed at earlier time-points when compared to *generic* MN preparations. MN cultures enriched in phMNs could not only be used to study phMN death in ALS but also, potentially, other respiratory MN diseases such as spinal muscular atrophy and Kennedy's Disease.

## Methods
### Human iPSC lines
Human iPSC line NCRM-1 was obtained from the National Institutes of Health Stem Cell Resource (Bethesda, MD, USA). Human iPSC lines CS29iALS-C9nxx (CS29-ALS hereafter for sake of brevity) and CS52iALS-C9nxx (CS52-ALS), corresponding to two distinct parental lines generated from ALS patients with *C9orf72* gene mutations, together with their matching isogenic controls CS29iALS-C9n1.ISOnxx (CS29 isogenic) and

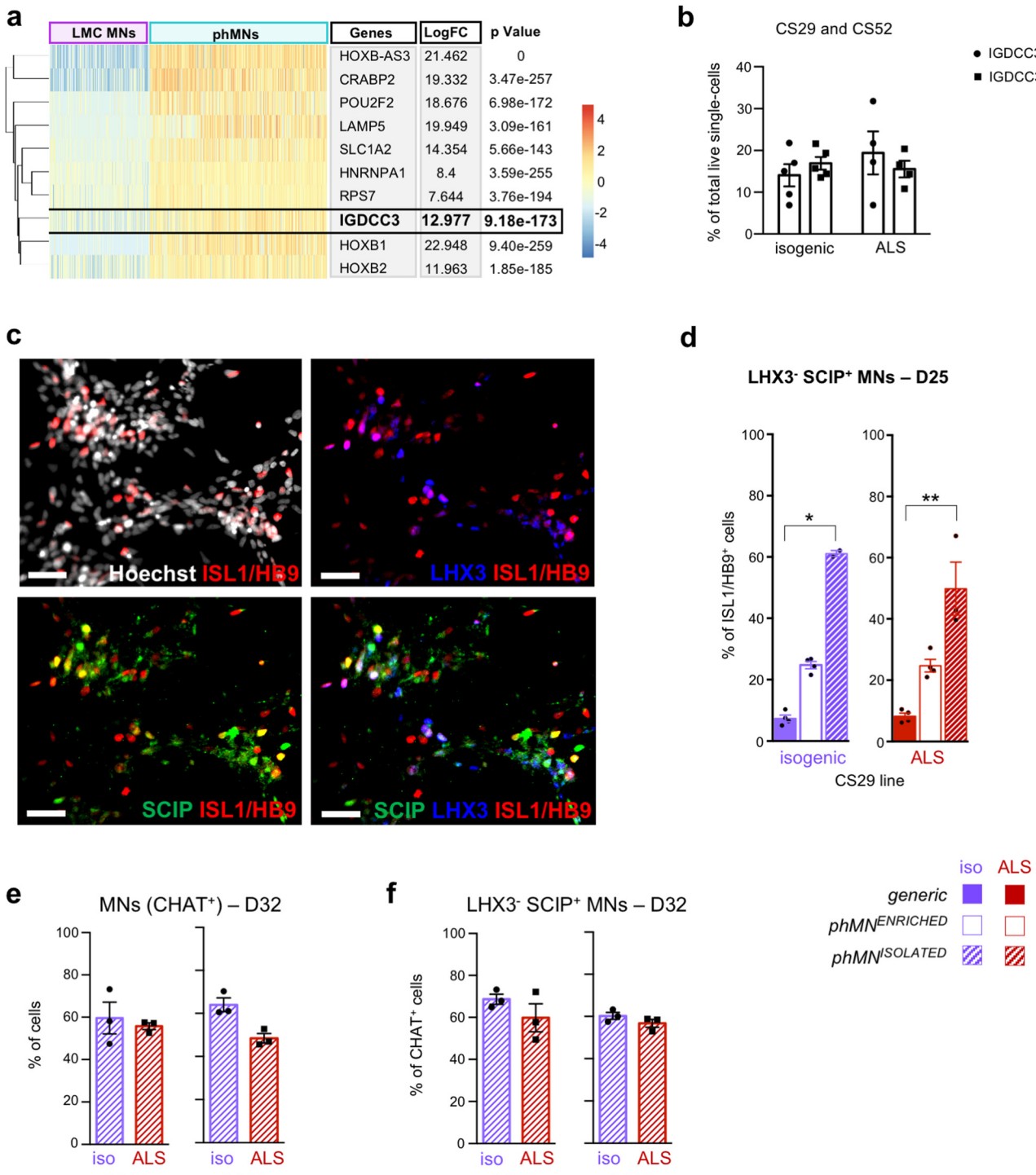

**Fig. 4 | Fluorescence Activated Cell Sorting (FACS) of phrenic motor neurons.**
**a** Heatmap of the 10 most differentially expressed genes (DEGs) identified by single-cell RNA sequencing as up-regulated in cluster #6 (corresponding to phMNs) compared to cluster #4 (corresponding to LMC MNs). Each bar corresponds to a cell and is coloured by the expression of the delineated gene relative to that in the other cluster. **b** Proportion of IGDCC3-positive and IGDCC3-negative cells isolated by flow cytometry from $phMN^{ENRICHED}$ cultures generated from CS29- and CS52 hiPSC lines (*C9orf72*-mutated (ALS) or isogenic). $N = 5$ biologically independent cultures for Isogenic, $N = 4$ biologically independent cultures for ALS. Error bars are means ± standard error of means (SEM) of the average. **c** Representative images of IGDCC3-positive cells, 3 days after FACS sorting. Cultures were co-stained for the pan-MN marker HB9/ISL1 (red), LHX3 (blue), and SCIP (green). Scale bars = 50 μm. **d** Quantification of phMNs, identified as ISL1$^+$/HB9$^+$ SCIP$^+$ LHX3$^-$, as percentages

of the number of ISL1$^+$/HB9$^+$ cells in *generic*, $phMN^{ENRICHED}$ or $phMN^{ISOLATED}$ cultures from CS29 Isogenic (purple) or ALS (red) iPSCs after 25 days of differentiation. Kruskal-Wallis non parametric test and Dunn's post-hoc multiple comparisons test; $* = p < 0.05$; $** = p < 0.005$; $N = 4$ biologically independent cultures for *generic* and $phMN^{ENRICHED}$; $N = 3$ biologically independent cultures for $phMN^{ISOLATED}$ (with > 500 cells in random fields for each culture). Error bars are means ± standard error of means (SEM) of the average. **e, f** Quantification of MNs identified as CHAT$^+$ cells, as percentages of the total number of cells (**e**), and phMNs identified as CHAT$^+$/LHX3$^-$/SCIP$^+$, as percentages of the number of CHAT$^+$ cells (**f**), two-weeks after FACS sorting of $phMN^{ENRICHED}$ cultures from ALS CS52 iPSCs (D32). Wilcoxon signed rank test; no significant difference; $N = 3$ biologically independent cultures (with > 500 cells in random fields for each culture). Error bars are means ± standard error of means (SEM) of the average. LMC Lumbar motor column.

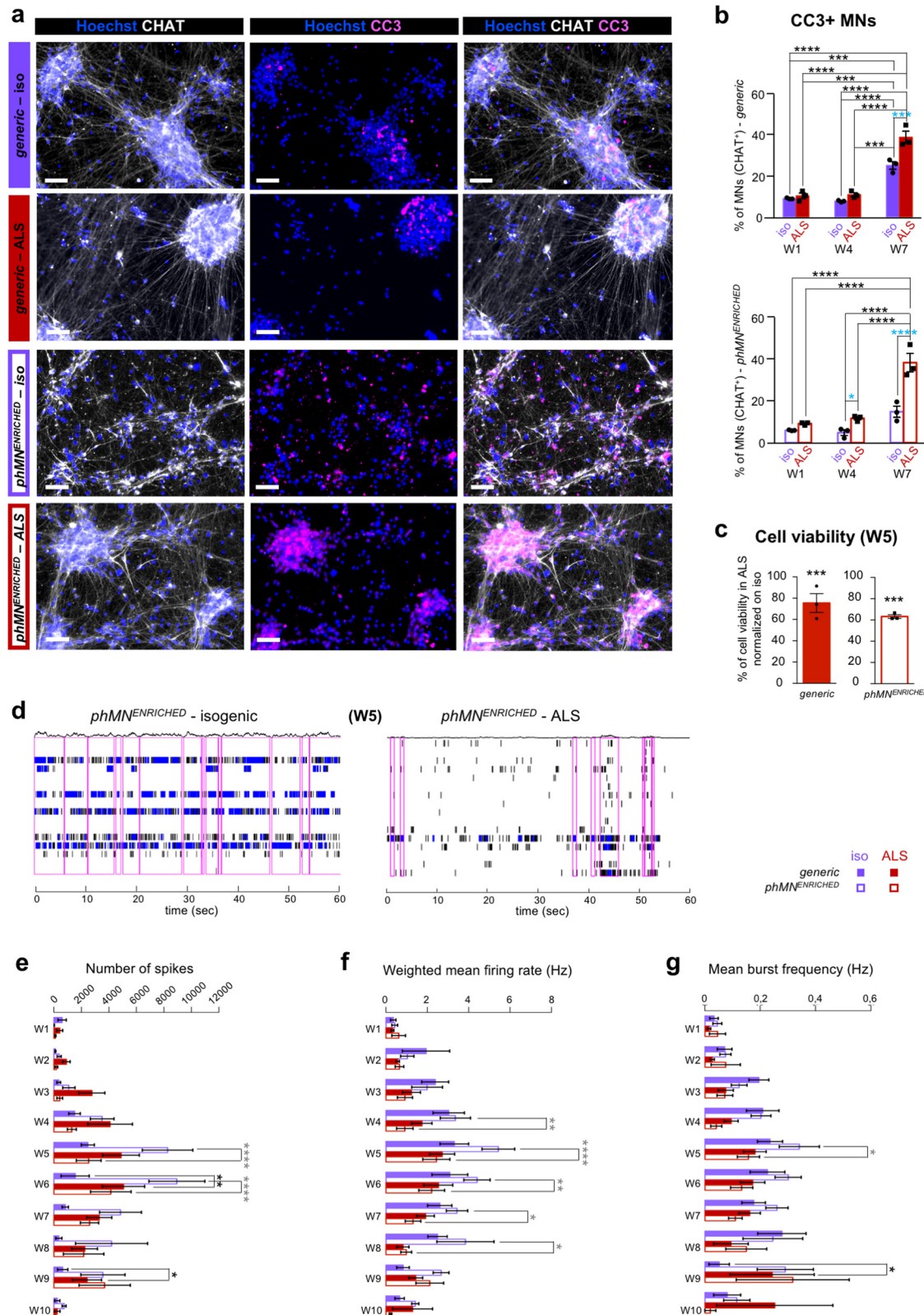

CS52iALS-C9n6.ISOnxx (CS52 isogenic), were obtained from Cedars-Sinai (Los Angeles, CA, USA)[32,35,63].

### Generation of motor neurons from hiPSCs

Human iPSCs were differentiated into NPCs, as described previously[22]. NPCs were then differentiated into MNPs, and subsequently MNs, using

either the standard protocol described previously[22], or 4 modified experimental conditions using different combination of varying concentrations of RA (Sigma-Aldrich; Oakville, ON, Canada; Cat. No. R2625) and purmorphamine (Pur.) (Sigma-Aldrich; Cat. No. SML-0868), as follow: 1) 0.1 μM RA + 0.5 μM Pur. ($RA^{LOW}/Pur.^{HIGH}$, defined as *generic* condition considering its use in numerous previous hiPSC-MN differentiation

**Fig. 5 | Effect of *C9orf72* mutation on cell death and activity of enriched phrenic motor neuron cultures. a** Representative images of MN cultures derived from Isogenic (purple) and *C9orf72*-mutated (ALS; red) CS29 iPSCs and stained with anti-CHAT (white) and anti-cleaved caspase 3 (CC3; magenta) antibodies after 53 days of differentiation (i.e., four-weeks (W4) post-plating) in each of the two experimental conditions: *generic* (filled boxes) or *phMN^ENRICHED* (hollow boxes). Scale bars = 50 μm.
**b** Quantification of CC3⁺/CHAT⁺ MNs as percentages of the total number of MNs (e.g., all CHAT⁺ cells) in *generic* (left, filled bars) or *phMN^ENRICHED* (right, hollow bars) cultures derived from the CS29 hiPSC line. Two-way ANOVA and Tukey's post-hoc multiple comparisons test; * = $p < 0.05$; ** = $p < 0.005$; *** = $p < 0.0005$; $N = 3$ biologically independent cultures per condition. Error bars are means ± standard error of means (SEM) of the average. Blue asterisks show statistical significance between Iso and ALS cells at the same time point. **c** Cell viability assay measured using CellTiter-GLO after 60 days of differentiation (i.e., five-weeks (W5) post-plating) in *generic* (left, filled bars) or *phMN^ENRICHED* (right, hollow bars) cultures derived from the CS29 hiPSC line. Cell viability for ALS cells was normalized on each respective corresponding isogenic control. Wilcoxon signed rank test; *** = $p < 0.0001$; $N = 3$ biologically independent cultures per condition. Error bars are means ± standard error of means (SEM) of the average. **d** Multielectrode array recording assays. Raster plots of the detected spikes on

each electrode within one random active well, representing one minute of spontaneous activity recorded from *phMN^ENRICHED* cultures after 60 days of differentiation (i.e., five-weeks (W5) post-plating) from isogenic (left) or ALS (right) CS29 hiPSCs. Each tick indicates the time a spike occurred and each row indicates an electrode. Blue ticks indicate the spikes that are part of a single-electrode burst. Magenta rectangles outline ticks that are included in network bursts (e.g., bursts detected as near synchronous spiking across electrodes in a well). Above each raster plot is a filtered population spike time histogram, showing the total number of spikes occurring throughout the well as a function of time. **e–g** Histograms showing the average number of spikes (**e**), weighted mean firing rate (Hz) (**f**), and mean burst frequency (Hz) (**g**) over time, from one-week to 10-weeks post-plating of isogenic *generic*, isogenic *phMN^ENRICHED*, ALS *generic*, and ALS *phMN^ENRICHED* from the CS29 hiPSC line. Two-way repeated measure ANOVA and Tukey's post-hoc multiple comparisons test; * = $p < 0.05$; ** = $p < 0.005$; *** = $p < 0.0005$; **** = $p < 0.0001$; $N = 5$ (W1 to W4), $N = 6$ (W5 to W7), $N = 3$ (W8 and W10), $N = 6$ (W9) biologically independent cultures per condition. Error bars are means ± standard error of means (SEM) of the average. Black asterisk = statistical significance for isogenic vs ALS *generic* cultures; Grey asterisk = statistical significance for isogenic vs ALS *phMN^ENRICHED* cultures.

studies [18,21,22,32,35]; 2) 1 μM RA + 0.125 μM Pur. (*RA^HIGH /Pur.^LOW*, defined as *phMN^ENRICHED* because it yielded cultures enriched in phMN-like cells); 3) 0.1 μM RA + 0.5 μM Pur. (*increased RA*); and 4) 0.1 μM RA + 0.125 μM Pur. (*reduced Pur.*). Briefly, NPCs were dissociated with Gentle Cell Dissociation Reagent (STEMCELL Technologies; Cat. No. 07174) on day 6 and split 1:5 with a chemically defined neural medium including DMEM/F12 supplemented with GlutaMAX (1/1; Thermo-Fisher Scientific; Cat. No. 35050-061), Neurobasal medium (1/1; Thermo-Fisher Scientific; Cat. No. 21103-049), N2 (0.5X; Thermo-Fisher Scientific; Cat. No. 17504-044), B27 (0.5X; Thermo-Fisher Scientific; Cat. No. 17502-048), ascorbic acid (100 μM; Sigma-Aldrich; Cat. No. A5960), antibiotic-antimycotic (1X; Thermo-Fisher Scientific; Cat. No. 15240-062), supplemented with RA (0.1 or 1 μM) and purmorphamine (0.5 or 0.125 μM) in combination with 1 μM CHIR99021 (STEMCELL Technologies; Cat. No. 72054), 2 μM DMH1 (Sigma-Aldrich; Cat. No. D8946) and 2 μM SB431542 (Tocris Bioscience; Bristol, UK; Cat. No. 1614). The culture medium was changed every other day for 6 days and the resulting MNPs were characterized by immunocytochemistry. MNPs were expanded for 6 days with the same medium containing 3 μM CHIR99021, 2 μM DMH1, 2 μM SB431542, 0.1 or 1 μM RA, 0.5 or 0.125 μM purmorphamine and 500 μM valproic acid (VPA; Sigma-Aldrich; Cat. No. P4543). To generate MNs, MNPCs were dissociated and plated at 50,000 cells/well on coverslips coated with dendritic polyglycerol amine (dPGA; 25 μg/mL; kindly provided by Dr. T.E. Kennedy) and Matrigel, as described previously[63]. MNs were cultured with the same neural medium supplemented with 0.5 or 1 μM RA and 0.05 μM or 0.1 μM purmorphamine, 0.1 Compound E (Calbiochem; Cat. No. 565790), insulin-like growth factor 1 (10 ng/mL; R&D Systems; Minneapolis, MN; Cat. No. 291-G1-200), brain-derived neurotrophic factor (10 ng/mL; Thermo-Fisher Scientific; Cat. No. PHC7074) and ciliary neurotrophic factor (10 ng/mL; R&D Systems; Cat. No. 257-NT-050). Culture medium was replaced every other day for 6 days and the resulting MNs were characterized by immunocytochemistry. For single cell RNA sequencing, MNs were dissociated, plated on T25 flasks coated with Matrigel, and cultured for 6 before single cell suspension preparation[22].

## Characterization of hiPSC-derived cells by immunocytochemistry

Induced MNPs and MNs were analyzed by immunocytochemistry, which was performed as described previously[64]. The following primary antibodies were used: mouse anti-OLIG2 (1/100; Millipore Corp.; Billerica, MA, USA; Cat. No. MABN50); goat anti-OLIG2 (1/200; R&D Systems; Cat. No. AF2418); rabbit anti-PAX6 (1/500; Covance; Emeryville, CA, USA; Cat. No. PRB-278P); mouse anti-PAX6 (1/150; Millipore Corp.; Cat. No. MAB5554); rat panTLE antibody (1/10)[65]; rabbit anti-HOMEOBOX A5 (HOXA5) (1/150; kindly provided by Dr. Jeremy Dasen, New York University School

of Medicine, New York, NY); mouse anti-NK2 HOMEOBOX 2 (NKX2.2) (1/100; DSHB; Cat. No. 74.5A5-c); mouse anti-HOMEOBOX PROTEIN HB9 (HB9) (1/30; DSHB; Cat. No. 81.5C10-c); mouse anti-ISLET1 (ISL1) (1/30; DSHB; Cat. No. 39.4D5-c); rabbit anti-LIMB HOMEOBOX CONTAINING 3 (LHX3) (1/100; Abcam; Toronto, ON, Canada; Cat. No. ab14555); goat anti-FORKHEAD BOX PROTEIN 1 (FOXP1) (1/100; R&D Systems; Cat. No. AF4534); mouse anti-neurofilament protein (2H3) (1/35; DSHB; Cat. No. 2H3-c); goat anti-CHAT (1/100; Millipore; Cat. No. MAB144P); and guinea pig anti-SCIP (1/16,000; kindly provided by Dr. Jeremy Dasen). Secondary antibodies against primary reagents raised in various species were conjugated to Alexa Fluor 350, Alexa Fluor 488, Alexa Fluor 555, or Alexa Fluor 647 (1/1000, Invitrogen; Burlington, ON, Canada). For quantification, images were acquired using an Axio Observer Z1 microscope connected to an AxioCam camera and using ZEN software (Zeiss). For each culture and each time point (e.g., NPCs, MNPs, MNs), images of > 500 cells in 3 random fields were taken with a 20X objective and analyzed with Image J.

## Preparation of single-cell suspensions from hiPSC-derived motor neuron cultures

Single-cells in suspension were prepared from MNs cultured for 28 days (Day 0 defined as start of NPC induction), as previously described[22]. Briefly, cells from one T-25 flask were dissociated with 2 mL of dissociation reagent containing papain (50U; Sigma-Aldrich; Cat. No. P4762) and Accutase (Thermo-Fischer Scientific; Cat. No. A11105-01). Cells in dissociation reagent were incubated for 15−20 min at 37 °C to ensure dissociation of cell clusters before adding 5 mL of DMEM/F12 containing 0.04% BSA and 10% ROCK inhibitor Y-27632 (1/1000; Tocris; Cat. No. 1254), followed by resuspension by gentle pipetting. Cells in suspension were transferred into a 15 mL falcon tube and centrifuged at 1300 rpm for 3 min at room temperature. The cell pellet was resuspended in a resuspension buffer containing 2 mL of dPBS (calcium and magnesium free phosphate buffered saline) and 0.04% BSA and ROCK-inhibitor (1/1000). Cells were centrifuged again at 1300 rpm for 3 min at room temperature and the cell pellet was resuspended in 500 μL of the above-mentioned resuspension buffer. A 30 μm strainer was used to remove cell debris and clumps and the single-cell suspension was transferred into a 2 mL Eppendorf tube placed on ice. Cells were counted to evaluate cell concentration and viability before single cell RNA sequencing. The targeted cell concentration was 700–1200 cells/μL, as recommended in the 10x Genomics guidelines (https://www.10xgenomics.com/solutions/single-cell/).

## Single cell RNA sequencing (sc-RNAseq) and in silico analysis

Cells were sequenced at a single-cell level using the microdroplet-based platform, 10x Genomics Chromium Single Cell 3' Solution (10x Genomics;

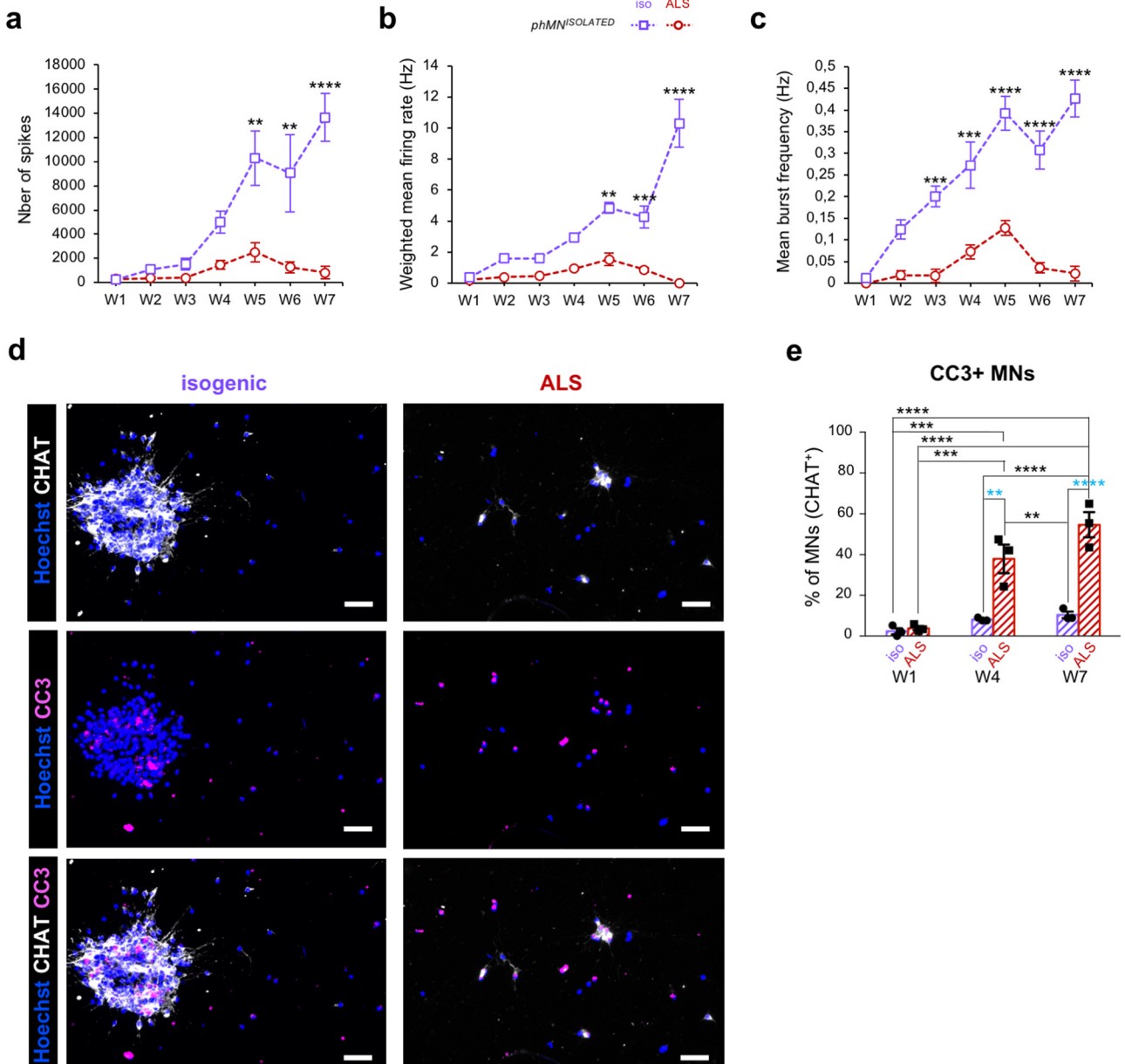

**Fig. 6 | Impact of _C9orf72_ mutation on cell death and activity in _isolated_ phrenic motor neuron cultures.** **a**–**c** Graph showing the average number of spikes (**a**), weighted mean firing rate (Hz) (**b**), and mean burst frequency (Hz) (**c**) over time, from one-week to 7-weeks post-plating of isogenic (purple line) and ALS (dotted line) _phMN_$^{ISOLATED}$ cultures derived from CS29 iPSCs. Two-way repeated measure ANOVA and Tukey's post-hoc multiple comparisons test; * = $p < 0.05$; ** = $p < 0.005$; *** = $p < 0.0005$; **** = $p < 0.0001$; $N = 3$ biologically independent cultures per condition. Error bars are means ± standard error of means (SEM) of the average. **d** Representative images of _phMN_$^{ISOLATED}$ cultures derived from isogenic (purple) and ALS (red) CS29 iPSC lines and stained with anti-CHAT (white) and anti-CC3 (magenta) antibodies four-weeks (W4) post-plating (i.e., 53 days of differentiation) of the same number of cells in both condition (i.e., 30,000 cells per coverslip). Scale bars = 50 μm. **e** Quantification of the CC3$^+$ MNs (e.g., CC3$^+$/CHAT$^+$ cells) as percentages of the total number of MNs (e.g., CHAT$^+$ cells) in _phMN_$^{ISOLATED}$ cultures derived from the CS29 iPSC line. Two-way repeated measure ANOVA and Tukey's post-hoc multiple comparisons test; * = $p < 0.05$; ** = $p < 0.005$; *** = $p < 0.0005$; **** = $p < 0.0001$; $N = 3$ biologically independent cultures per condition. Error bars are means ± standard error of means (SEM) of the average. Blue asterisks show statistical significance between Iso and ALS at the same time point.

Pleasanton, Ca, USA), followed by sequencing on a HiSeq4000 system (Illumina; San Diego, CA, USA) at the McGill and Genome Quebec Innovation Center (https://cesgq.com/en-services). The 10x cDNA libraries were sequenced at a depth of 50,000 reads per cell. The raw sc-RNAseq data (FASTQ files) were first processed using the Cell Ranger pipeline (10x Genomics) to demultiplex and align the sequences to the human reference genome, GRCh38. FASTQ files were aligned and empty droplet 10x barcodes were filtered out via Cell Ranger. Post-Cell Ranger, gene/cell matrices were imported into R and all the subsequent data analysis was conducted in R version 3.5.2 using Bioconductor Software Version 3.8. All software

packages are publicly available at the Bioconductor project (http://bioconductor.org).

### Fluorescence Activated Cell Sorting (FACS) of phrenic motor neurons

After 18 days of differentiation, _phMN_$^{ENRICHED}$ cultures obtained from isogenic or ALS-patient derived hiPSCs were dissociated into single-cell suspensions at approximately $1\text{-}10^6$ cells/mL as described above. Cell suspensions were washed twice and resuspended in 1 mL of Dulbecco's PBS (D-PBS) before viability staining using Live/Dead™ Fixable Aqua (1/1000;

ThermoFisher) according to manufacturer's protocol. The cell pellets were resuspended in FACS Buffer (5% FBS, 0.1% NaN3 in D-PBS) containing optimal concentration determined by antibody titration of the Alexa Fluor® 647-conjugated IGDCC3 extracellular antibody (1:20; R&D Systems; Cat. No. FAB8559R). After a 30 min incubation in the dark at room temperature, cells were washed twice with FACS Buffer and centrifuged at 350 g for 5 min Phrenic MNPs were isolated with FACS Aria Fusion (Becton-Dicksinson Biosciences), using a nozzle with a size of 20 μm (Becton Dickinson) and a sheath pressure of 20 pounds per square inch (psi). The AlexaFluor 647 molecule was excited using a 640 nm laser and captured with a bandpass filter of 670/30 of the 640nm-Detector C. The Fixable Aqua molecule was excited using a 405 nm laser and captured with a bandpass filter of 450/50 BP of the 405 nm-Detector F. Using an unstained control sample, a gate was drawn to select IGDCC3-positive cells and 15−20% of the higher IGDCC3-positive cells were sorted and collected into 1 mL of DMEM/F12 supplemented with neurotrophic factors. FACS-isolated cells ($phMN^{ISOLATED}$) were seeded at a density of 20,000 cells per well on 6-well plates coated with Matrigel and cultured for 6 days in DMEM/F12 supplemented with 3 μM CHIR99021, 2 μM DMH1, 2 μM SB431542, 0.1 or 1 μM RA, 0.5 or 0.125 μM purmorphamine and 500 μM VPA before generation of MNs as described above.

### Cell death assays of hiPSC derived motor neuron cultures
Two assays were used to assess cell death in hiPSC-derived MN cultures, as briefly described below:

**Quantification of apoptotic motor neurons by immunocytochemistry.** After 53 days of differentiation (i.e., 4-weeks post-plating), rabbit anti-Cleaved-Caspase 3 (CC3 (Asp175); 1/400; Cell Signaling; New England Biolabs, Ltd., Ontario, Canada; Cat. No. 9661S) was used in combination with goat anti-CHAT antibody (1/100; Millipore; Cat. No. MAB144P) to detect apoptotic MNs in cultures obtained from isogenic or ALS-patient derived cells from the CS52 or the CS29 line, in the *generic*, $phMN^{ENRICHED}$, and $phMN^{ISOLATED}$ conditions. For quantification, images in 3 random fields were taken with a 20X objective using an Axio Observer Z1 microscope connected to an AxioCam camera using ZEN software (Zeiss) and analyzed with Image J.

**CellTiter-Glo luminescent cell viability assay.** The CellTiter-Glo assay (Promega; Madison, Wisconsin, USA; Cat. No. G7570) was used according to the manufacturer's instructions to determine the proportion of viable cells in isogenic and ALS derived-cultures at four-weeks post-plating (i.e., 53 days of differentiation). For each culture condition (isogenic *generic*, ALS *generic*, isogenic $phMN^{ENRICHED}$ and ALS $phMN^{ENRICHED}$), an N of 3 cultures were studied. The cell viability assay was performed between one-week (control) and four-weeks post-plating (test day), and the luminescence was quantified by the GloMax Luminometer plate reader (Promega).

### Multi-electrode array recordings of hiPSC derived motor neuron cultures
After 24 days of differentiation, hiPSC-derived MNs were plated at ~50,000 cells/well on dPGA/Matrigel-coated wells of CytoView MEA 24-well plates (Axion BioSystems; Atlanta, Georgia, USA; Cat. No. M384-tMEA-24W). Recordings from 16 electrodes per well were conducted using a Maestro (Axion BioSystems) MEA recording amplifier with a head stage maintaining a temperature of 32˚C at 5% CO2. MEA plates were allowed to equilibrate for ~3 min prior to the 5 min recording of spontaneous activity. Data were sampled at 12.5 kHz, digitized, and analyzed using Axion Integrated Studio software (Axion BioSystems) with a band-pass filter (200–5000 Hz). The spontaneous electrophysiological activity of the hiPSC-derived cells was recorded every week post-plating for 10 weeks. The following electrophysiological parameters are presented in this study: number of spikes, firing rate, and burst frequency.

### Statistics and reproducibility
To account for culture variability and ensure experimental reproducibility, each experiment was performed with at least 3 biologically independent cultures per condition. Error bars shown are means ± standard error of means (SEM) of the average.

To detect differences in the proportion of the different cell types obtained from the NCRM1 hiPSC line in the four culture conditions, we used one-way ANOVA and Holm-Sidak's multiple comparisons post hoc test or Kruskal-Wallis nonparametric test and Dunn's multiple comparisons post hoc test when the variables did not fit a normal distribution (assessed by Kolmogorov-Smirnov test).

All statistical tests used to detect differences between isogenic and ALS patient-derived hiPSCs, in either the *generic* or the $phMN^{ENRICHED}$ culture conditions, were conducted separately for the two pairs of *C9orf72*-mutated hiPSC lines (CS29 and CS52). To detect differences in the proportion of the different cell types obtained from isogenic or ALS patient-derived cells in each of the two culture conditions (*generic* and $phMN^{ENRICHED}$), we used the t-test or the nonparametric Mann-Whitney ranked sum test when the variables did not fit a normal distribution (assessed by Kolmogorov-Smirnov test). The Wilcoxon signed rank test was used to detect differences in cell viability measured with the Cell Titer Glo assay in isogenic and ALS patient-derived cells after 60 days of differentiation in each of the two culture conditions, and cell viability for ALS cells was normalized on each respective corresponding isogenic control. Two-way repeated measure ANOVA and Tukey's multiple comparisons post-hoc test were used to detect differences in (1) the proportion of CC3-positive apoptotic MNs or (2) the cells' electrophysiological properties in isogenic and ALS patient-derived cells in each of the three culture conditions (generic, $phMN^{ENRICHED}$, or $phMN^{ISOLATED}$) at different time points.

### Reporting summary
Further information on research design is available in the Nature Portfolio Reporting Summary linked to this article.

## Data availability
Numerical source data behind each graph presented in this study are publicly available in. Figshare under the identifier (DOI): 10.6084/m9.figshare.24985722[66]. The raw sc-RNAseq data used in the present study is available at the Sequence Read Archive (SRA) http://www.ncbi.nlm.nih.gov/bioproject/1065451[67].

## Code availability
The R workflow used for the analysis was adapted from a published workflow used to characterize hiPSC-derived MN cultures[22].

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

## Acknowledgements

We thank Vincent Soubannier, Nisha Pulimood, Yeman Tang, and Rita Lo for discussions and advice. These studies were funded in part by grants to SS from the Canadian Institutes for Health Research and Fonds de la recherche en Sante-Quebec under the frame of E-Rare-3, the ERA-Net for Research on Rare Diseases. TMD was supported by the Canada First Research Excellence Fund, awarded through the HBHL initiative at McGill University, the CQDM's Health Collaborations Accelerator Fund program and a project grant from CIHR (PJT – 169095). SS is a Distinguished James McGill Professor of McGill University.

## Author contributions

L.T. designed and performed experiments, data analysis, figures and wrote the first draft of the manuscript. J.S. performed the FACS experiments. L.T., T.M.D., S.S. conceived the study plan. S.S. supervised data analysis and manuscript writing.

## Competing interests

The authors declare no competing interests.
