## [Peer Review File · Communications Biology]

Reviewers' comments:

Reviewer #1 (Remarks to the Author):

The manuscript by Thirty et al., investigates the molecular (patho)physiology of phrenic MNs (phMN) derived from hiPSCs from patients carrying the C9orf72 amplification, one of the most common amyotrophic lateral sclerosis (ALS) mutations. ALS patients are known to die of respiratory failure; however, current disease models lack the specificity to study the molecular mechanisms underlying phrenic MN degeneration. The authors generate a human iPSC phrenic MN model that they then utilize to investigate neuronal cell death and changes in activity in C9orf72 hiPSC-phMNs. The authors identify specific concentrations of retinoic acid (RA) and the SHH agonist purmorphamine that lead to an increase in MN progenitors (MNPs) with a dorsal/cervical identity and an enrichment in phrenic MNs in their cultures, as compared to standard RA/SHH treatments. However, more experimental support is needed for some of the conclusions and interpretations proposed by the authors, including additional validation of their model. My main questions/concerns are listed below:

1. The modified RA/SHH concentrations used by the authors lead to about 15% dorsal and 15% Hoxa5+ (cervical) MNPs. Presumably the overlap of both pathways leads to phrenic MNs but the authors do not show quantitation of dorsal and cervical marker overlap. It is also not clear what the authors are calling TLE low in the images or what the threshold was for quantifying dorsal-most progenitors. An image of TLE staining alone or more clarification in the methods would be useful.
2. In figure 2 the representative images shown do not correspond to the data quantification shown in the graphs. For example, MNs are shown to be about 70% in all conditions in the graphs but in the images the red (Hb9/Isl+) cells look to be less than 50%. This is also inconsistent with the single-cell sequencing data that show only 44% and 34% of cells to be MNs in the generic and phrenic-enriched conditions respectively. Similarly, Lhx3+ cells appear to be way more than 10%. Also the definition of phrenic MNs as Scip+/Lhx3- is incomplete. Do these cells express Hoxa5? Are they FoxP1-negative (especially since they appear to be generated at the expense of FoxP1+ neurons)? Do they express other markers known to be phrenic MN-specific (see Machado et al. 2014)? Additional validation of what is defined as phrenic MNs is needed.
3. The use of IGDC3 as a phrenic-specific marker is inadequately justified. While this comes out of the single-cell sequencing data, the expression pattern of this marker in vivo is not shown. Is it specific to phrenic MNs in vivo? Other differential markers shown on the list are Hoxb1 and Hoxb2 which are not expressed in phrenic MNs in vivo, reducing confidence in the approach used to choose this cell surface marker. Additional validation is needed to show that this protein is in fact phrenic-specific. After using IGDC3 to FACS MNs, 60% are considered to be phrenic MNs because they are Lhx3- and Scip+. What about Hoxa5 and FoxP1 expression? Again, more validation is needed.
4. In fig 5A, the CC3+ cells look similar in both conditions, unlike what is shown in the quantitation. In 5E it seems that the difference is a result of the isogenic enriched cultures having a much higher activity than the generic, rather than a reduced activity of the ALS lines between the two conditions (generic and enriched). Can the authors comment on this more extensively? What % of each culture plated on the MEAs is expected to be phrenic MNs in the 2 conditions? Based on the single cell data, 34% of cells are MNs in the enriched condition and less than 50% are classified as phrenic MNs. Are MNs sorted before being plated on the MEA?
5. While the need for pure MN populations to study disease mechanisms is clear, the claim by the authors that phrenic MN selectively degenerate in MN diseases such as ALS is not well supported by current experimental evidence. The authors might want to tone this claim down and provide

alternative justification for the study.

6. The authors refer to phrenic MNs as a subset of the HMC. This is not entirely accurate, as there are substantial differences between the two MN populations, justifying classification of phrenic MNs as a distinct column.

Reviewer #2 (Remarks to the Author):

This paper describes a method that enables in the derivation of phrenic-like MNs from human iPSCs. Currently, our understanding of the mechanisms of phMN degradation in ALS is limited, mainly due to the lack of human experimental models to study phMNs, whereas they provide a method for generating human phMNs using an optimized combination of small molecules followed by cell sorting based on cell surface proteins enriched in hiPSC- phMNs, these studies establish a previously unavailable generation of human phMNs, providing a disease-relevant system for studying the mechanisms of respiratory MN dysfunction. This work is important and I think the paper will be useful to a wide audience and is suitable for publication. However, some corrections are still needed.

1. Whether the advantages and disadvantages of the new and old models should be compared, and whether the stability of the model is guaranteed.
2. The identification of cells is very comprehensive, but there is a lack of functional studies, and the follow-up research is worth looking forward to.
3. May the color matching of your figure could be optimized.
4. Why is Sonic hedgehog signal chosen for this experiment?
5. Please elaborate on Retinoic Acid 's relevance to this experiment and the reasons for choosing it.
6. Please add experimental evidence that the use of human ipsc derivatives rich in diaphragmatic motor neurons can mimic respiratory MN death disease in ALS patients.
7. Why choose purmorphamine as Sonic hedgehog inhibitor?

RE: Resubmission of Manuscript COMMSBIO-23-2625-T

Dear Dr. Akhtar:

Thank you for your letter of 6 October 2023, regarding our submission COMMSBIO-23-2625-T. We are delighted to submit a revised version that has been modified to address the comments made by the Reviewers. To facilitate the re-review process, we have uploaded a marked-up version of our manuscript in which textual revisions are highlighted in yellow; all new text is also underlined.

Below you will find a response to all points raised by the two Referees who assessed our manuscript. Please note that we have indicated the manuscript lines where modifications to the text were made (hopefully line numbering will not change during pdf generation at submission). We have also underlined all new figures in this rebuttal.

Reviewer #1

1. a) The modified RA/SHH concentrations used by the authors lead to about 15% dorsal and 15% Hoxa5+ (cervical) MNPs. Presumably the overlap of both pathways leads to phrenic MNs but the authors do not show quantitation of dorsal and cervical marker overlap.

We agree that this is an important point. Since we needed to reserve one fluorescence channel for nuclear counterstaining, in order to quantify the percent of immunostained cells present in each culture, we were not able to use 4 separate markers in our original studies. Moreover, we could not test the combination of PAX6, TLE, and HOXA5, because we used a rabbit anti-PAX6 antibody (anti-HOXA5 antibody was also raised in rabbit). As a result, we did not assess the coexpression of dorsal-most ($PAX6^{HIGH}/TLE^{LOW}$) and cervical ($HOXA5$) markers.

To address the reviewer's comment to the best of our ability, we have conducted new ICC experiments in which quadruple labelling (without Hoechst staining) was made possible by new anti-PAX6 (mouse) and anti-OLIG2 (goat) antibodies (please see lines 391-394 and 404 in the revised manuscript text for antibody details) together with anti-TLE (rat) and anti-HOXA5 (rabbit) antibodies. These experiments demonstrated the presence of cells with the $TLE^{LOW}/PAX6^{HIGH}/HOXA5^{+}$ profile indicative of cervical dorsal-most MN progenitors, as requested (please see Supplementary Figure 2C; to facilitate re-review, we have embedded these new data below). The text describing these new results can be found on lines 104-105: "Most of $HOXA5^{+}$ MNPs were also $PAX6^{HIGH}/TLE^{LOW}$ ("dorsal-most"), confirming the specification of dorsal MNPs with cervical identity (Supplementary Figure 2C)."

Supplementary Figure 2. Dorsalization and caudalization of motor progenitor cultures. [...] (C) Representative images of MNP cultures stained with anti-HOXA5 (white), anti-PAX6 (red), anti-TLE (green) and anti-OLIG2 (blue) antibodies after 12 days of differentiation in the *generic* and the $RA^{HIGH}/Pur.^{LOW}$ conditions. Scale bars = 50 μ m.

1. b) It is also not clear what the authors are calling TLE low in the images or what the threshold was for quantifying dorsal-most progenitors. An image of TLE staining alone or more clarification in the methods would be useful.

We thank the Reviewer for requesting this important clarification. The finding that dorsal-most MNPs at cervical and thoracic spinal cord levels *in vivo* can be identified through the combination of PAX6 and TLE expression (eg, $PAX6^{HIGH}/TLE^{LOW}$) was originally made in the developing spinal cord by Salin-Cantegrel *et al.* (2020; doi:10.1101/2020.03.10.986323). To provide further evidence that varying levels of PAX6 and TLE expression also define different MNP cell populations in hiPSC-derived preparations, we have now added new data showing that 1) the characterization of TLE^{HIGH} vs TLE^{LOW} and $PAX6^{HIGH}$ vs $PAX6^{LOW}$ expression provides a means to identify different subtypes of hiPSC-derived MNPs, and 2) dorsal-most (eg, $PAX6^{HIGH}/TLE^{LOW}$) MNPs are enriched under the experimental condition that generates increased amounts of SCIP+ MNs. Please see Supplementary Figure 1 (to facilitate re-review, we have embedded these new data below).

We also edited the text to clarify the process used for identification of $PAX6^{HIGH}/TLE^{LOW}$ cells, as follows: “Analysis of the relative levels of fluorescence intensity of TLE and PAX6 staining in each cell allowed for the identification of their $PAX6^{HIGH}/TLE^{LOW}$ (“dorsal-most”) expression profile (Supplementary Figure 1).” Please see lines 86-89.

Supplementary Figure 1. Identification of PAX6^{HIGH}/TLE^{LOW} MNP cells. (A) Representative images of MNP cultures stained with anti-PAX6 (red) and anti-TLE (green) antibodies after 12 days of differentiation under each of the four experimental conditions defined in Figure 1A. Scale bars = 50 μ m. (B) Plot of the fluorescence intensity of PAX6 and TLE staining for each cell (e.g., region of interest (ROI)) present in each representative image presented in (A). Cells that are PAX6^{LOW}/TLE^{HIGH} are represented in the green rectangular area and correspond mostly to cells of the *generic* condition (black squares). In contrast, cells that are PAX6^{HIGH}/TLE^{LOW} are represented in the red rectangular area and correspond mostly to cells of the *RA^{HIGH}/Pur.^{LOW}* condition (hollow black squares).

2. a) In figure 2 the representative images shown do not correspond to the data quantification shown in the graphs. For example, MNs are shown to be about 70% in all conditions in the graphs but in the images the red (Hb9/Isl+) cells look to be less than 50%. This is also inconsistent with the single-cell sequencing data that show only 44% and 34% of cells to be MNs in the generic and phrenic-enriched conditions respectively. Similarly, Lhx3+ cells appear to be way more than 10%.

We have edited Figure 2A to provide images that better reflect the quantification data, showing that approximately 60-70% of cells are MNs, and that no more than 10% of cells are LHX3-positive MNs. For better visualization, we have used blue to show LHX3 staining and green for FOXP1 or SCIP staining. To facilitate re-review, we have embedded these new data below.

Regarding the point about the inconsistency between ICC results (60-70% MNs) and sc-RNaseq results (35-40% MNs), we respectfully wish to note that we had addressed this situation in the

Discussion of our first submission. Regardless, we have now strengthened the text as follows: “Of note, the relative proportion of induced phMNs appeared lower in scRNA-seq studies compared to ICC. This could reflect a specific loss of phMNs over other MN types during single-cell suspension preparation prior to sequencing. Mature hiPSC-derived MNs tend to coalesce into large clusters^{17,32,48,54–57}, and phMNs present a distinctive tight clustering very early during in vivo development^{58,59}. The presence of these clusters necessitates harsh conditions to obtain single-cell suspensions for sc-RNAseq²², and it is possible that phMNs are particularly sensitive to these conditions” (please see lines 286-292).

Figure 2. Characterization of enriched human iPSC-derived phrenic motor neurons by immunocytochemistry. (A) Representative images of motor neurons (MNs) co-stained with antibodies against the known pan-MN marker HB9/ISL1 (red), LHX3 (blue), and FOXP1 or SCIP (green) after 25 days of differentiation under the four experimental conditions defined in Figure 1A. Arrows indicate phMNs, identified as HB9/ISL1⁺ LHX3⁻ SCIP⁺.

2. b) The definition of phrenic MNs as Scip+/Lhx3- is incomplete. Do these cells express Hoxa5? Are they FoxP1-negative (especially since they appear to be generated at the expense of FoxP1+ neurons)? Do they express other markers known to be phrenic MN-specific (see Machado et al. 2014)? Additional validation of what is defined as phrenic MNs is needed.

Since we could not combine more than 4 markers in ICC studies without the risk of losing accuracy, phMNs were identified as ISL1⁺/HB9⁺/SCIP⁺/LHX3⁻ in combination with Hoechst staining for quantitative analysis. It was our belief that the sc-RNAseq we presented were more informative than ICC to define phMNs, thus overcoming the technical constraints inherent to ICC. Moreover, sc-RNAseq allowed us to analyse and compare the transcriptomics profiles for phMN and other MNs.

Regardless, we have done our best to address the reviewer’s comment by adding new data showing additional validation of phMNs with the co-staining of ISL1/HB9 with SCIP and HOXA5, or with SCIP

and FOXP1 (please see Supplementary Figure 3; to facilitate re-review, we have embedded these new data below). To text describing these new results can be found on lines 122-123: “SCIP⁺ phMNs were also HOXA5⁺ and FOXP1-negative (Supplementary Figure 3), further defining their phMN molecular identity^{7,8}.”

Supplementary Figure 3. SCIP⁺ phrenic motor neurons express HOXA5 but not FOXP1. (A) Representative images of MNs co-stained with pan-MN marker HB9/ISL1 (blue), anti-HOXA5 (red), and anti-SCIP (green) antibodies after 25 days of culture in the *generic* and the $RA^{HIGH}/Pur.^{LOW}$ conditions. (B) Representative images of MNs co-stained with anti-HB9/ISL1 (blue), anti-FOXP1 (red), and anti-SCIP (green) antibodies after 25 days of culture in the *generic* and the $RA^{HIGH}/Pur.^{LOW}$ conditions. Scale bars = 50 μ m.

3. a) The use of IGDCC3 as a phrenic-specific marker is inadequately justified. While this comes out of the single-cell sequencing data, the expression pattern of this marker in vivo is not shown. Is it specific to phrenic MNs in vivo? Other differential markers shown on the list are Hoxb1 and Hoxb2 which are not expressed in phrenic MNs in vivo, reducing confidence in the approach used to choose this cell surface marker. Additional validation is needed to show that this protein is in fact phrenic-specific.

We thank the reviewer for raising this important point. We apologise if our text was not clear enough. We did not intend to say that IGDCC3 was specific to phMNs. What we intended to say was that scRNAseq analysis of the two main MN populations found in iPSC-derived phMN^{ENRICHED} cultures showed that IGDCC3 is enriched in the phMN-cluster as compared to the LMC-cluster. Based on this

observation, we demonstrated that *IGDCC3* can be used successfully to enrich iPSC-derived preparations for cells corresponding to the phMN-cluster.

We have rephrased the text as follows to better explain these points: “*IGDCC3* is expressed in the mouse developing nervous system, including spinal MNs. In the spinal cord, *IGDCC3* is down-regulated in maturing MNs^{60,61}. Thus, although *IGDCC3* expression is not limited to phMNs, its expression in different MN subtypes might have different dynamics resulting in an apparent up-regulation in phMNs compared to LMC at a given time in development.” (please see text on lines 297-301).

3.b) After using *IGDCC3* to FACS MNs, 60% are considered to be phrenic MNs because they are *Lhx3*- and *Scip*+. What about *Hoxa5* and *FoxP1* expression? Again, more validation is needed.

To address this comment, we have added new data providing additional validation of phMNs in post-FACS cultures through co-staining of CHAT, SCIP and HOXA5 or CHAT, SCIP and FOXP1 (please see Supplementary Figure 6; to facilitate re-review, we have embedded these new data below.) The text describing these results can be found on lines 193-195: “Post-FACS SCIP⁺ cells were also FOXP1-negative and HOXA5⁺, further confirming their phMN molecular identity.”

Supplementary Figure 6. Post-FACS SCIP⁺ phrenic motor neurons express HOXA5 but not FOXP1. Representative images of MNs co-stained with anti-CHAT (blue), anti-SCIP (green) antibodies, and either anti-HOXA5 antibody (red, **A**) or anti-FOXP1 antibody (red, **B**), two-weeks after FACS sorting of phMN^{ENRICHED} cultures from ALS CS52 iPSCs (D32). Scale bars = 50 μ m.

4. a) In fig 5A, the CC3⁺ cells look similar in both conditions, unlike what is shown in the quantitation.

We edited Figure 5A to provide the best possible representative image for CC3⁺ cells in *generic* ALS cultures, illustrating the fact that at four-weeks (W4) there is no statistically significant difference in the proportion of apoptotic MNs between Isogenic and ALS *generic* MNs. This is explained in the text (lines 211-216) as follows: “Significant difference between isogenic and ALS MNs was only detected after 7-weeks of maturation in the *generic* condition (Figure 5A and B) (although not statistically significant in the case of the CS52 line (Supplementary Figure 7A)). In contrast, when cells were

differentiated using the $phMN^{ENRICHED}$ condition, the proportion of dying MNs was higher in ALS compared to isogenic cultures already at 4 weeks post-plating, with both the CS29 (Figure 5A and B) and CS52 (Supplementary Figure 7A) lines.”

Figure 5. Effect of *C9orf72* mutation on cell death and activity of enriched phrenic motor neuron cultures. (A) Representative images of MN cultures derived from Isogenic (purple) and *C9orf72*-mutated (ALS; red) CS29 iPSCs and stained with anti-CHAT (white) and anti-cleaved caspase 3 (CC3; magenta) antibodies after 53 days of differentiation (i.e., four-weeks (W4) post-plating) in each of the two experimental conditions: *generic* (filled boxes) or $phMN^{ENRICHED}$ (hollow boxes). Scale bars = 50 μ m. (B) Quantification of CC3⁺/CHAT⁺ MNs as percentages of the total number of MNs (e.g., all CHAT⁺ cells) in *generic* (left, filled bars) or $phMN^{ENRICHED}$ (right, hollow bars) cultures derived from the CS29 hiPSC line. Two-way ANOVA and Tukey’s post-hoc multiple comparisons test; *= $p < 0.05$; **= $p < 0.005$; ***= $p < 0.0005$; N= 3 cultures per condition. Blue asterisks show statistical significance between Iso and ALS cells at the same time point. (C) Cell viability assay measured using CellTiter-GLO after 60 days of differentiation (i.e., five-weeks (W5) post-plating) in *generic* (left, filled bars) or $phMN^{ENRICHED}$ (right, hollow bars) cultures derived from the CS29 hiPSC line. Cell viability for ALS cells was normalized on each respective corresponding isogenic control. Wilcoxon signed rank test; ***= $p < 0.0001$; N= 3 cultures per condition. [...]

4. b) In 5E it seems that the difference is a result of the isogenic enriched cultures having a much higher activity than the generic, rather than a reduced activity of the ALS lines between the two conditions (generic and enriched). Can the authors comment on this more extensively? What % of each culture plated on the MEAs is expected to be phrenic MNs in the 2 conditions? Based on the single cell data, 34% of cells are MNs in the enriched condition and less than 50% are classified as phrenic MNs. Are MNs sorted before being plated on the MEA?

We thank the reviewer for this important comment. Our data show that isogenic *enriched* cultures have a higher activity than *generic* cultures, while ALS *enriched* and *generic* preparations show relatively similar levels of activity. Compared to isogenic *enriched* cultures, ALS *enriched* cultures have a significantly lower activity: this phenotype could not be observed using *generic* cultures. We have

better explained this in the text as follows: “The latter observation could be explained, at least in part, by the finding that isogenic *phMN*^{ENRICHED} cultures were significantly more active than isogenic *generic* preparations, thereby facilitating the detection of loss of activity in ALS cultures enriched in phMNs. The intrinsic hyperactivity of isogenic cultures enriched in phMNs, compared to *generic* preparations, may reflect the specific ability of phMNs to spontaneously fire very early during *in vivo* development⁵⁹, thus further supporting the physiological relevance of hiPSC-derived phMN preparations to study excitability changes in respiratory MN diseases.” (please see lines 331-337).

5. While the need for pure MN populations to study disease mechanisms is clear, the claim by the authors that phrenic MN selectively degenerate in MN diseases such as ALS is not well supported by current experimental evidence. The authors might want to tone this claim down and provide alternative justification for the study.

As suggested by both reviewers, we have edited the text to clarify the rationale for using phrenic MNs to study respiratory MN death in ALS, including new references (please see lines 28-38). The revised text now reads: “Respiratory failure is the primary cause of death in ALS patients^{1,2} and survival time is significantly shorter for patients with respiratory-onset ALS than bulbar- or limb-onset ALS^{3,4}, highlighting the importance of respiratory muscles in ALS patient survival and the crucial need to precisely understand the mechanisms underlying the respiratory pathology of the disease. The loss of the respiratory function of ALS patients is mainly the result of the degeneration of phMNs that control the major inspiratory muscle, the diaphragm⁵. Phrenic MNs are different from other spinal MNs in a number of ways, including their distinct developmental origin, topology, and electrophysiological properties⁶⁻⁹. Moreover, phMN progressive loss in ALS animal models begins before symptoms onset¹⁰, and is more pronounced than other spinal respiratory MNs, such as hypoglossal (XII) MNs^{11,12}, reflecting the recognized selectivity of cell death mechanisms among different MN subtypes^{5,13-16}.”

6. The authors refer to phrenic MNs as a subset of the HMC. This is not entirely accurate, as there are substantial differences between the two MN populations, justifying classification of phrenic MNs as a distinct column.

We thank the reviewer for this important comment. As suggested by the reviewer, we have edited the text throughout the paper to refer to phrenic MNs as a distinct motor column. Figures 3 and 4 were also edited to reflect these specific changes.

Reviewer #2

1. Whether the advantages and disadvantages of the new and old models should be compared, and whether the stability of the model is guaranteed.

We have rephrased the text of the Discussion as follows: “Our results demonstrate that these preparations offer a disease-relevant model system to study the mechanisms underlying the respiratory pathology of ALS. Specifically, these cultures can be studied for at least 10 weeks and disease-related phenotypes including cell death and loss of electrophysiological activity are consistently observed at earlier time-points when compared to *generic* MN preparations. MN cultures enriched in phMNs could not only be used to study phMN death in ALS but also, potentially, other respiratory MN diseases such as spinal muscular atrophy and Kennedy’s Disease.” (please see lines 338-345).

2. The identification of cells is very comprehensive, but there is a lack of functional studies, and the follow-up research is worth looking forward to.

The Editor indicated that the request for further functional studies is not mandatory for further consideration at this journal.

3. May the color matching of your figure could be optimized.

The color code for each condition throughout the figures paper was optimized as described in the following table. This table is only submitted here for the reviewers' consideration. We also updated the graphs of Figure 6A-C to match these color edits.

Experimental condition	control (NCRM1)	Isogenic	ALS
generic	filled, black	filled, purple	filled, red
$RA^{HIGH}/Pur.^{LOW}$ phMN ^{REDUCED}	= hollow, black	hollow, purple	hollow, red
increased RA	hollow, grey	NA	NA
reduced Pur.	filled, grey	NA	NA
phMN ^{ISOLATED}	NA	dashed, purple	dashed, red

4. Why is Sonic hedgehog signal chosen for this experiment?

As described in the Introduction, a dorsal to ventral graded Sonic hedgehog (SHH) signaling is implicated in the specification of developing motor neurons (MNs). As the spinal cord develops, phrenic MNs are generated from dorsal-most MN progenitors (MNPs) in response to this graded SHH signaling, whereas postural MNs of the median motor column (MMC) derive from more ventral MNPs (Jessell 2000). As first shown in the seminal study by the late Tom Jessell's group in 2002 (Wichterle, et al. Cell 2002. DOI: 10.1016/s0092-8674(02)00835-8), specific combinations of small molecules can be used to generate spinal MNs from hiPSCs by mimicking *in vitro* the sequential steps of *in vivo* MN development. Specifically, SHH signaling is activated by adding to the culture media the SHH agonist purmorphamine at defined concentrations, leading to the ventralization of neural progenitor cells to obtain (MNPs) (Wichterle et al., 2002; Li et al., 2005; Amoroso et al., 2013; Du et al., 2015, and many more publications). We followed the established use of purmorphamine as a SSH agonist: more importantly, we hypothesized that reducing SHH signaling would promote the generation of dorsal-most MNPs, from which phrenic MNs arise. These concepts are explained in the manuscript text, together with the rationale for using Retinoic Acid (in response to comment #5).

Please see the following text on lines 50-59: "During spinal cord development, phMNs emerge from specific MN progenitors (MNPs) located in the 'dorsal-most' MN progenitor (pMN) domain at cervical level ^{7,8,23,24}. Specification of the pMN domain is under the control of a ventral to dorsal gradient of Sonic hedgehog (SHH) signaling emanating from the notochord and floor plate ²⁵⁻²⁷. Cervical identity is controlled by a rostro-caudal gradient of Retinoic Acid (RA), which regulates HOXA5 gene expression in the cervical segment of the spinal cord, contributing to phMN identity specification ^{7,8,24}. Thus, we hypothesized that a calibrated activation of SHH and RA signaling in hiPSC-derived neural progenitor cells (NPCs) might provide a strategy to enhance the specification of dorsal MNPs with cervical identity, in turn generating neurons with phMNs features."

5. Please elaborate on Retinoic Acid 's relevance to this experiment and the reasons for choosing it.

As mentioned in the introduction, the positional identity of MNs along the rostro-caudal axis of the spinal cord is determined by a rostro-caudal gradient of Retinoic Acid (RA) signaling (Liu et al., 2001; Dasen & Jessell, 2009). High levels of RA determine HOXA5 gene expression in the cervical segment of the spinal cord, thus promoting rostral (e.g. cervical) identity and contributing to phMN specification (Philippidou et al., 2012; Machado et al., 2014). We hypothesized that increasing RA signaling would promote the generation of cervical MNPs, from which phMNs arise. As mentioned above (in response to point #4), we edited the text (lines 50-59) to provide a better explanation for the method used to generate iPSC-derived MNs cultures enriched in phrenic MNs.

6. Please add experimental evidence that the use of human ipsc derivatives rich in diaphragmatic motor neurons can mimic respiratory MN death disease in ALS patients.

The present studies already describe several experimental results providing evidence that human-iPSC derived cultures enriched in phrenic MNs better are a better model for respiratory MN death in ALS compared to *generic* MN cultures. First, Figures 5A and 5B show that the number of apoptotic MNs (CC3⁺/CHAT⁺) is increased in ALS MNs compared to isogenic MNs as early as 4-weeks post-plating using *enriched* cultures, while cell death is not observed in *generic* cultures before 7-weeks. Second, Figures 5D to 5G show that isogenic cultures *enriched* in phMNs present a significantly higher electrophysiological activity compared to isogenic *generic* cultures: this situation enabled us to detect a difference between isogenic and ALS MNs that could not be observed using *generic* cultures, thereby offering an improved experimental system. These two phenotypes (increased death and decreased electrical activity of ALS MNs) are even more pronounced upon greater enrichment of phrenic MNs following FACS (Figure 6), further strengthening the conclusion that cultures enriched in phMNs provide an advanced model system to study mechanisms of respiratory MN death.

7. Why choose purmorphamine as Sonic hedgehog inhibitor?

As mentioned above, the use of Purmorphamine as a SHH agonist was originally described in the seminal work by the Jessell's group (Wichterle, et al. Cell 2002. DOI: 10.1016/s0092-8674(02)00835-8). Since then, Purmorphamine has been used in many studies as a SHH signaling agonist to generate spinal motor neurons from hiPSCs (eg, Li et al., 2008; Amoroso et al., 2013; Du et al., 2015, to list only very few out of many).

In conclusion, we believe that by adding a considerable amount of new data and rephrasing/improving the text in several section, our resubmission has adequately addressed all the concerns of both Reviewers. We are looking forward to a positive response.

Sincerely yours

Stefano Stifani, PhD
Distinguished James McGill Professor of Neurology and Neurosurgery, McGill University
Associate Director (Research), Montreal Neurological Institute-Hospital

REVIEWERS' COMMENTS:

Reviewer #1 (Remarks to the Author):

The authors addressed all of my previous concerns and I think the manuscript has been improved by inclusion of additional data and quantitation.

Reviewer #2 (Remarks to the Author):

The author addressed my comments carefully. I think they are novel and will be of interest to others. I indeed meet some problems in ALS model. Might be the manuscript inspired me.